# Tunable phenotypic variability through an autoregulatory alternative sigma factor circuit

Christian P Schwall[1,†], Torkel E Loman[1,†], Bruno M C Martins[1,2], Sandra Cortijo[1], Casandra Villava[1], Vassili Kusmartsev[1], Toby Livesey[1], Teresa Saez[1] & James C W Locke[1,*]

## Abstract

Genetically identical individuals in bacterial populations can display significant phenotypic variability. This variability can be functional, for example by allowing a fraction of stress prepared cells to survive an otherwise lethal stress. The optimal fraction of stress prepared cells depends on environmental conditions. However, how bacterial populations modulate their level of phenotypic variability remains unclear. Here we show that the alternative sigma factor $\sigma^V$ circuit in *Bacillus subtilis* generates functional phenotypic variability that can be tuned by stress level, environmental history and genetic perturbations. Using single-cell time-lapse microscopy and microfluidics, we find the fraction of cells that immediately activate $\sigma^V$ under lysozyme stress depends on stress level and on a transcriptional memory of previous stress. Iteration between model and experiment reveals that this tunability can be explained by the autoregulatory feedback structure of the *sigV* operon. As predicted by the model, genetic perturbations to the operon also modulate the response variability. The conserved sigma-anti-sigma autoregulation motif is thus a simple mechanism for bacterial populations to modulate their heterogeneity based on their environment.

**Keywords** *Bacillus subtilis*; microbial systems biology; single-cell time-lapse microscopy; stochastic gene expression; stress priming
**Subject Category** Microbiology, Virology & Host Pathogen Interaction
**Mol Syst Biol. (2021) 17: e9832**

## Introduction

Cells live in a changeable environment and experience a wide range of environmental stresses. Bacterial populations have evolved strategies to survive these stresses. One strategy is for all cells to immediately respond to stress with the activation of the relevant stress response pathway (Hilker *et al*, 2016). Alternatively, a bacterial population can generate a broad range of cellular states, which allows it to hedge its bets against the changeable environment

(Veening *et al*, 2008b). Noise in gene expression has been proposed as a mechanism for generating phenotypic variability in genetically identical cells (Raj & van Oudenaarden, 2008; Martins & Locke, 2015). This phenotypic variability has also been shown to be affected by changes in the cellular environment, such as a shift in stress level or growth conditions (Megerle *et al*, 2008; de Jong *et al*, 2012; Mitosch *et al*, 2019), as well as by previous 'priming' stresses (Mitosch *et al*, 2017). However, how the bacterial population regulates individual cell decisions to modulate the fraction of cells that enter an alternative transcriptional state remains unclear (Fig 1A).

The $\sigma^V$ mediated lysozyme stress response pathway in *Bacillus subtilis* is an ideal model system to examine how bacterial populations can tune their phenotypic variability. $\sigma^V$ is an extracytoplasmic function (ECF) alternative sigma factor. Alternative sigma factors replace the 'housekeeping' sigma factor, $\sigma^A$, in the RNA polymerase holoenzyme and redirect it to regulons that control distinct regulatory programmes. They have already been shown to display a high level of gene expression variability in *B. subtilis* (Locke *et al*, 2011; Young *et al*, 2013; Cabeen *et al*, 2017; Park *et al*, 2018), and the $\sigma^V$ activation pathway is both well characterized and specific to one stress condition, which greatly simplifies analysis of its activation.

$\sigma^V$ is the only pathway activated in response to C-type lysozyme (Guariglia-Oropeza & Helmann, 2011; Ho *et al*, 2011; Ho & Ellermeier, 2012). Lysozyme is produced by animals as part of their innate immune system and kills bacteria by cleaving the peptidoglycan of the cell wall between the *N*-acetylmuramic acid residue and the β-(1,4)-linked *N*-acetylglucosamine (Lal & Caplan, 2011). In its inactive form, $\sigma^V$ is bound to its anti-sigma factor RsiV, a transmembrane protein (Fig 1B). If lysozyme is present, RsiV binds to lysozyme (Hastie *et al*, 2014; Hastie *et al*, 2016) and activates a signal transduction cascade to release $\sigma^V$. First RsiV undergoes a conformational change that allows signalling peptidases to cleave RsiV at site-1 (Hastie *et al*, 2014; Castro *et al*, 2018; Lewerke *et al*, 2018) (Fig 1B). *Bacillus subtilis* has five type 1 signal peptidases, of which the two major peptidases are SipS and SipT (Tjalsma *et al*, 1998). Either SipS or SipT is sufficient for site-1 cleavage (Castro *et al*, 2018; Ho & Ellermeier, 2019). The truncated RsiV can then be cleaved by RasP (a site-2 protease), which results in the release of $\sigma^V$ (Hastie *et al*, 2013; Hastie *et al*, 2014) (Fig 1B).

---

1 Sainsbury Laboratory, University of Cambridge, Cambridge, UK
2 School of Life Sciences, University of Warwick, Coventry, UK
*Corresponding author. Tel: +44 1223 761110; E-mail: james.locke@slcu.cam.ac.uk
†These authors contributed equally to this work

Once σ[V] is released it can bind to RNA polymerase and redirect transcription to the σ[V] regulon. The regulon contains genes that allow adaptation to lysozyme stress, such as the penicillin-binding protein PbpX and the D-alanyl-D-alanine carrier protein ligase DltA. Both these genes are also part of other sigma factor pathways (Guariglia-Oropeza & Helmann, 2011; Ho *et al*, 2011). Like many alternative sigma factors, one of σ[V]'s targets is its own operon. The *sigV* operon includes *sigV* (the gene that codes for σ[V]) and its anti-sigma factor *rsiV*, and so its activation results in 'mixed' positive and negative feedback loops. In addition, the operon contains *oatA*, one of the main lysozyme resistance genes and *yrhK*, which is of unknown function (Hastie & Ellermeier, 2016). OatA contributes to lysozyme adaptation by transferring an acetyl group to the C-6-OH

position of *N*-acetylmuramic acid in the peptidoglycan, thus blocking cell wall cleavage by lysozyme (Bernard *et al*, 2011; Ho *et al*, 2011).

Although the molecular mechanisms underlying σ[V] activation by lysozyme have been elucidated, previous work was carried out using bulk experiments that average out single-cell dynamics (Guariglia-Oropeza & Helmann, 2011; Ho *et al*, 2011) and mask cell-to-cell heterogeneity. This heterogeneity can be crucial in identifying and distinguishing between regulatory strategies (Munsky *et al*, 2012). In this study, we used single-cell time-lapse microscopy of fluorescent reporters for σ[V] activity to characterize σ[V] induction dynamics in individual cells in response to lysozyme stress. We found that upon induction by sub-lethal levels of lysozyme, σ[V] is

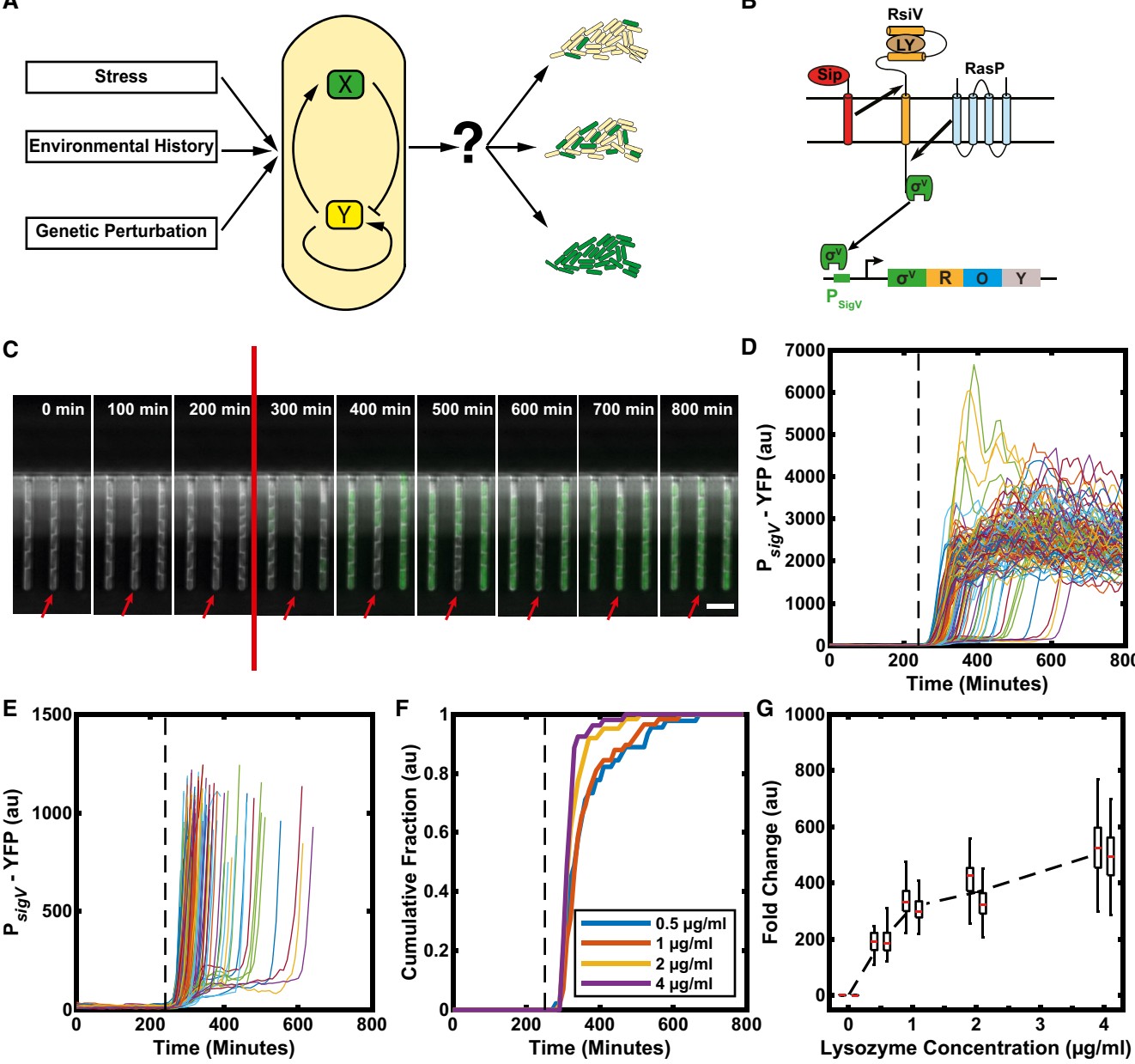

Figure 1.

**Figure 1.  σ^V is activated heterogeneously in response to lysozyme stress.**

A   It is unclear how genetic circuits (centre) tune phenotypic variability in a bacterial population in response to genetic perturbation, environmental stress and history (left) to modulate the fraction of cells that activate a given pathway (right).

B   Schematic of the σ^V circuit. In its inactive form, σ^V is bound to its anti-sigma factor RsiV. If there is lysozyme present in the environment, RsiV binds to lysozyme and undergoes a conformational change. Only then the proteases SipS/ SipT and RasP can cleave RsiV to release σ^V. Once σ^V is released from RsiV, it can bind to RNA polymerase to redirect transcription to the σ^V regulon. σ^V can initiate transcription of its own operon which includes *sigV* (σ^V), *rsiV* (R), *oatA* (O, one of the main lysozyme resistance genes) and *yrhK* (Y, unknown function).

C   Time-lapse microscopy of cells containing a P*sigV*-YFP promoter reporter reveals heterogeneous activation of σ^V in response to lysozyme stress. The stress (red line) was added between the 200 and 300 min time points (at 240 min). The red arrow highlights a cell with a delayed activation of σ^V. Scale bar: 5 μm.

D   Time traces of mean YFP fluorescence per cell. Each trace represents a single-cell lineage's response to 1 μg/ml lysozyme. (109 traces from two independent experiments). Lysozyme added at the black dashed line.

E   Figure D replotted to allow examination of variable activation times.

F   The observed heterogeneity is reduced with increasing stress levels. Each line represents the cumulative fraction of cells ($N = \sim 50$) with P*sigV*-YFP values that are higher than the half maximum of their final values (representing cells that have activated).

G   The fold change in mean YFP fluorescence increases with increasing stress levels, shown for two biological repeats (adjacent boxplots). Each day's fold change distribution consisted of > 40 manually corrected single-cell traces. On each box, the central mark indicates the median, and the bottom and top edges of the box indicate the 25$^{th}$ and 75$^{th}$ percentiles, respectively. The lower and higher whiskers of boxplot are extended to the first quartile minus 1.5 * interquartile range and the third quartile plus 1.5 * interquartile range, respectively. The black dashed line is the mean fold change.

Data information: For more information on the number of repeats, please see Appendix Table S3.

activated heterogeneously, with some cells activating σ^V rapidly, whereas some cell lineages do not activate σ^V for multiple generations. This heterogeneity is functional, as cells that respond to an initial sub-lethal stress are more likely to survive a subsequent lethal stress of lysozyme. Through experiment and modelling, we found that these dynamics can be explained solely by the autoregulatory feedback of σ^V on its own operon. Our model predicted that the observed heterogeneity could be tuned by environmental history and by genetic perturbations, which we confirmed experimentally. The conserved sigma-anti-sigma autoregulation motif is thus a simple mechanism for bacterial populations to tune their levels of phenotypic variability.

# Results

## Individual cells activate σ^V heterogeneously under lysozyme stress

To characterize σ^V activation dynamics, we first constructed a *B. subtilis* reporter strain containing a chromosomally integrated fluorescent reporter for σ^V activity, P*sigV*-YFP. This strain also contained a reporter for the housekeeping sigma factor σ^A (P*trpE*-RFP). P*trpE*-RFP expression was used as a constitutive control and to aid image analysis (Materials and Methods). We then used time-lapse microscopy to examine σ^V activity in individual cells grown in the mother machine microfluidic device (Wang *et al*, 2010). This device allows long-term tracking of hundreds of individual cells trapped at the ends of channels. It also allows for fast switching between media conditions during imaging (Materials and Methods). We first measured σ^V expression dynamics in response to a step change to a sub-lethal concentration of 1 μg/ml lysozyme (Fig 1C and D, Appendix Fig S1 and Fig EV1 and Movie EV1). The first cells begin to respond to the lysozyme stress with raised P*sigV*-YFP levels within two frames (20 min), suggesting rapid activation given the ~ 15 min maturation time of the YFP reporter protein (Appendix Fig S2). However, our time-lapse movies revealed that P*sigV*-YFP was heterogeneously activated at the single-cell level (Fig 1C). While 20% of cells reached the half maximum of σ^V activity within 70 min of being exposed to lysozyme, it took 200 min (approximately four

generations) for 90% of all cells to activate σ^V (Fig 1D and Appendix Fig S3B).

To test whether the observed heterogeneous activation of σ^V in response to lysozyme was due to the growth conditions in the mother machine, we investigated σ^V activation dynamics in liquid culture (Appendix Fig S4) and using an alternative microfluidic device (Appendix Fig S5). Both conditions showed similar heterogeneous σ^V activation to that observed in the mother machine. Therefore, the observed heterogeneity in P*sigV*-YFP is independent of the experimental setup. Previous work has shown that heterogeneous gene expression could be due to intrinsic noise in the expression of the P*sigV*-YFP reporter (Elowitz *et al*, 2002), rather than due to heterogeneous σ^V activity. To test whether the observed heterogeneity was due to the intrinsic variability of the P*sigV*-YFP reporter, we constructed a strain containing chromosomally integrated P*sigV*-YFP and P*sigV*-mTurq reporters and exposed it to 1 μg/ml lysozyme in liquid culture. In these snapshot experiments, the expression of P*sigV*-YFP and P*sigV*-mTurq was highly correlated ($R^2 = 0.85$) (Appendix Fig S6). Thus, the observed heterogeneity in P*sigV*-YFP reflects changes in σ^V activity and not intrinsic variability of the P*sigV*-YFP promoter.

Next, we asked whether the observed heterogeneity in P*sigV*-YFP is modulated by the level of lysozyme applied. We examined P*sigV*-YFP expression after the application of 0.5, 1, 2 and 4 μg/ml lysozyme. These values were all below the previously reported minimal growth inhibitory concentration of 6.25 μg/ml (Ho *et al*, 2011) and led to a transient decrease in growth rate (Appendix Fig S7). To measure the distribution of σ^V activation times, for each time point we calculated the fraction of cells that had crossed the half-maximum of their respective final σ^V value (meaning the cell had activated σ^V). We found that when increasing the lysozyme concentration from 0.5 to 4 μg/ml the heterogeneity in σ^V activity was reduced (Figs 1F and EV1 and Appendix Fig S8). The time for 90% of cells to activate their σ^V pathway decreased from 300 min (approximately six generations) for 0.5 μg/ml to 100 min (approximately two generations) for 4 μg/ml lysozyme (Appendix Fig S3). At the same time, the fold change in induction between the unstressed σ^V activity and the steady-state σ^V activity under lysozyme stress increased from ~ 190 for 0.5 μg/ml to ~ 520 for 4 μg/ml (Fig 1G and Appendix Fig S1). We also observed that under a 4 μg/ml

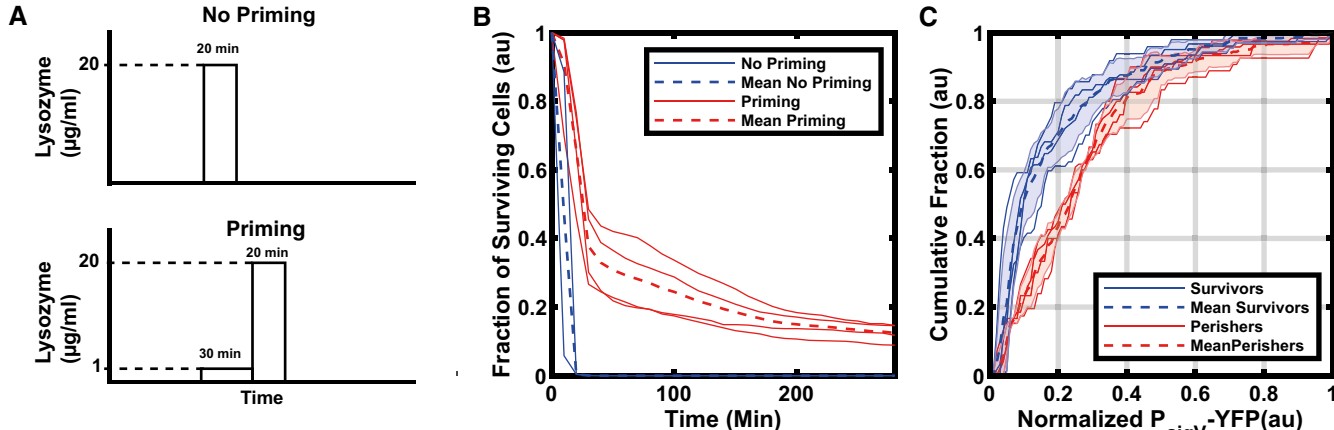

**Figure 2. Rapid activation of σ^V after a first stress application increases survival after a second higher stress.**

A  Schematic of lysozyme application. Cells are either exposed directly to a high concentration of lysozyme (20 μg/ml) for 20 min (top) or exposed first to a short (30 min) lysozyme priming stress (1 μg/ml) before exposure to the higher concentration (bottom).

B  A priming (30 min) stress of 1 μg/ml lysozyme followed by the high lysozyme stress (20 μg/ml) improves survival. The solid blue lines are the biological repeats for the no priming experiment ($n = 2$) whereas the solid red lines are the biological repeats for the priming experiment ($n = 4$). (Total number of cells shown for No Priming, $N = 2,013$ and Priming, $N = 4,937$).

C  Surviving cells have higher $P_{sigV}$-YFP levels after the initial priming stress (1 μg/ml) than perishing cells. The cumulative distributions were normalized by their maximum σ^V activity and baseline subtracted. Each dashed line is the mean of experiments from $n = 4$ biological repeats. The shaded areas correspond to the mean $\pm$ s.d. For more information on the number of repeats and cell numbers, please see the supplementary text.

Data information: For more information on the number of repeats, please see Appendix Table S3.

concentration of lysozyme some cells (8 and 21% in two different repeat experiments) appeared sick and were wider than usual cells. These cells also overshot their new σ^V activity steady-state before relaxing to it. We removed these cells from our analysis (Appendix Fig S9), although including them did not affect our results (Appendix Fig S10). We also observed that the fraction of cells activating σ^V increased with increasing lysozyme in the alternative microfluidic device, although in this device movies were stopped before all cells activated due to crowding of the cells in the chip (Appendix Fig S5F). Taken together, our results reveal that the level of phenotypic variability in σ^V activation times is tuned by stress levels, with the heterogeneity in σ^V activation reduced as lysozyme levels increase.

We examined whether the differences in individual cell states before applying a lysozyme stress could predict how a cell responds to the stress. There was no correlation between either growth rate or cell size before lysozyme application and response time after lysozyme application (Appendix Figs S11 and S12). Alternative sigma factors in *B. subtilis* display gene expression variability even in the absence of stress (Locke *et al*, 2011; Park *et al*, 2018). We found that, in the absence of stress, the coefficient of variation of $P_{sigV}$-YFP was over four times higher than that of the constitutive control ($0.65 \pm 0.29$ vs $0.14 \pm 0.05$) (Appendix Fig S13). Cells which had elevated levels of $P_{sigV}$-YFP before the addition of stress were more likely to activate σ^V instantaneously (Fig EV2 and Appendix Fig S14). However, some cells with low $P_{sigV}$-YFP levels also activate σ^V instantaneously (Fig EV2 and Appendix Fig S14). Therefore, while noise in $P_{sigV}$-YFP expression before stress contributes to the observed heterogeneity in σ^V activation after the addition of lysozyme, it is not the only cause.

Given the heterogeneity in σ^V activation times, we examined whether activating σ^V early had any effect on the survival against future lethal concentrations of lysozyme. Cells that were exposed to 20 μg/ml of lysozyme for 20 min all died within 1 h (Fig 2 and Appendix Fig S15). However, if the cells were first exposed to a priming stress of 1 μg/ml of lysozyme for 30 min, which heterogeneously induced σ^V, and then subsequently to 20 μg/ml of lysozyme for 20 min, some cells survived the high lysozyme stress (Fig 2A). We chose this priming stress level and duration as previous experiments had shown (Appendix Fig S4) that it ensured heterogeneous activation of σ^V, with a large fraction of cells not having turned on before the second higher stress. We found that the short exposure to sub-lethal concentrations of lysozyme (during the priming stress) improved survival to subsequent lethal stress levels from 0 to $12.5 \pm 2.7\%$ (See Materials and Methods, Fig 2B and Appendix Fig S15). Cells that survived until the end of the movie, 280 min after the 20 μg/ml of lysozyme, had on average $1.57 \pm 0.33$ fold higher $P_{sigV}$-YFP levels than perishing cells immediately before the application of the lethal concentration of lysozyme (Fig 2C).

### The heterogeneous activation of $P_{sigV}$-YFP is solely due to σ^V and its anti-sigma factor RsiV

We next attempted to understand how the single-cell activation dynamics of the σ^V pathway are generated. First, to test whether the heterogeneity that we observed in σ^V activation times was due to lysozyme stress activating different stress response pathways, we analysed the genome-wide transcriptional response of cells to 1 μg/ml lysozyme. We carried out RNA-seq 30 min after the addition of stress in the wild type and in the Δ*sigV* knockout. We chose this

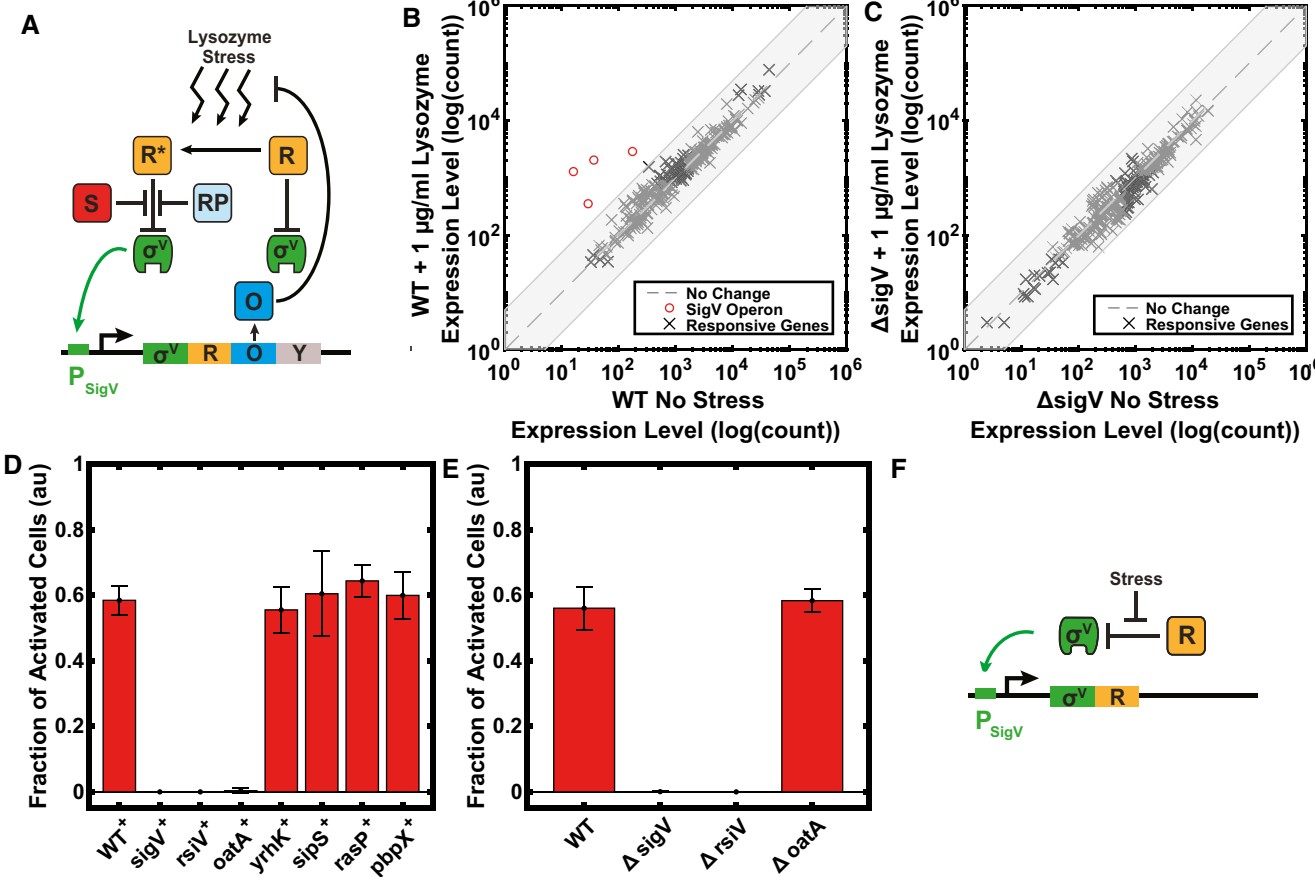

**Figure 3. The observed σ^V heterogeneity can be explained by a simplified σ^V circuit.**

A    Schematic of the σ^V circuit. In the figure, R (orange) is the anti-sigma factor RsiV, R* (orange) is RsiV bound to lysozyme, S (red) is signalling peptidase, RP (light blue) is RasP the site-2 protease, O (blue) is OatA, and Y (grey) is YrhK. For more information on the activation mechanism, see Fig 1B.

B, C   RNA-seq experiment on WT (JLB130) and Δ*sigV* (JLB154) strains, showing quantification of the absolute expression of individual genes in the presence and absence of lysozyme stress. The shaded grey box represents a ± 5 fold change. DEseq was used to identify genes which were differentially expressed with a 5% *P*-value cut-off between the WT and Δ*sigV* mutant in response to lysozyme treatment (default DEseq test was used). (B) Only the *sigV* operon is strongly activated (> 5 fold change) in response to lysozyme stress in WT (JLB130), as previously reported (Guariglia-Oropeza & Helmann, 2011). (C) No genes were strongly upregulated in Δ*sigV* (JLB154) by lysozyme stress.

D    Effect of the overexpression of individual components of the σ^V pathway (Biological repeats: WT: $n = 8$, *sigV*+: $n = 4$, *rsiV*+: $n = 3$ *oatA*+: $n = 4$, *yrhK*+: $n = 3$, *sipS*+: $n = 3$ and *rasP*+: $n = 3$, *pbpX*+: $n = 6$) on the fraction of activated cells. Only overexpression of *sigV*, *rsiV* or *oatA* changed the observed dynamics compared to WT. The histograms of the shown data are shown in Appendix Fig S16.

E    Deleting *oatA* did not alter the σ^V activation dynamics. However, deleting *sigV* or *rsiV* resulted in no further activation of σ^V in response to 1 ug/ml lysozyme. $n \geq 3$ biological repeats for all data shown. The histograms of the shown data are shown in Appendix Fig S17.

F    Schematic of simplified σ^V circuit with only σ^V and RsiV, where σ^V (green) activates its own expression and that of its anti-sigma RsiV (orange, R).

Data information: Bars correspond to the mean ± s.d. For more information on the number of repeats, please see Appendix Table S3.

stress level and duration as we had seen in previous experiments that it resulted in heterogeneous σ^V activation (Appendix Fig S4). As previously reported, only the *sigV* operon was strongly (> 5 fold induction) induced by lysozyme (Guariglia-Oropeza & Helmann, 2011) in the wild type (Fig 3B). The lysozyme resistance gene *pbpX*, which is part of the *sigV* regulon, was also upregulated in the WT (4.9 fold induction), but was not upregulated in the Δ*sigV* background (0.88 fold induction), consistent with the known role of *sigV* in its activation (Guariglia-Oropeza & Helmann, 2011). The Δ*sigV* strain did not show any genes with strong (> 5 fold induction) induction (Fig 3C). DltA, which is known to contribute to lysozyme resistance and is part of the *sigV* regulon (Guariglia-Oropeza &

Helmann, 2011), was not upregulated in the WT or the Δ*sigV* strain. Our results suggest that no other pathway is strongly induced by 1 μg/ml lysozyme. We therefore hypothesized that the observed P_*sigV*-YFP heterogeneity is due to the σ^V circuit itself (Ho *et al*, 2011; Hastie *et al*, 2013; Hastie *et al*, 2014).

To test which components of the σ^V circuit play a role in the heterogeneous activation of σ^V, we constructed IPTG-inducible over-expression constructs of all genes in the *sigV* operon (*sigV, rsiV, oatA, yrhK*), as well as genes in the σ^V circuit (*sipS, rasP*) and the *sigV* regulon (*pbpX*) in a P_*sigV*-YFP reporter background (Fig 3A). We then investigated σ^V activation dynamics under lysozyme stress after full induction of each circuit component. To do this, we took

single-cell snapshots of P$_{sigV}$-YFP expression of cells grown in liquid culture. The cells were either unstressed or were stressed for 30 min with 1 µg/ml lysozyme. As a metric for the variability in activation time of σ$^V$ in response to lysozyme, we calculated the fraction of cells with P$_{sigV}$-YFP levels larger than a threshold above the unstressed level of each strain (See Materials and Methods). Single-cell snapshots of P$_{sigV}$-YFP expression revealed that overexpressing either *yrhK*, *sipS*, *rasP* or *pbpX* only had a minor effect on the fraction of activated σ$^V$ cells (Fig 3D) or on the level of P$_{sigV}$-YFP expression (Appendix Fig S16). For cells overexpressing *sigV* itself, no further activation of σ$^V$ occurred on the addition of lysozyme (Fig 3 D). This was because P$_{sigV}$-YFP expression was already at a high level prior to stress (Appendix Fig S16). Conversely, overexpression of *rsiV* caused cells not to express P$_{sigV}$-YFP at all, preventing activation of all cells on addition of lysozyme (Fig 3D and Appendix Fig S16). Finally, overexpressing *oatA*, which blocks lysozyme cleavage of the peptidoglycan, shuts off the activation of σ$^V$ (Fig 3D and Appendix Fig S16). However, when we increased the lysozyme concentration to 20 µg/ml in the presence of *oatA* overexpression, the heterogeneous expression of P$_{sigV}$-YFP reappeared (Fig EV3), suggesting *oatA* is not responsible for the heterogeneous activation dynamics. We repeated the experiment for *rsiV* overexpression, but increased RsiV did not increase protect against lysozyme as a concentration of 20 µg/ml lysozyme killed all cells.

To further validate the importance of *sigV* and *rsiV* as compared to *oatA* for the observed heterogeneity in σ$^V$ activation, we constructed deletion mutants of *sigV*, *rsiV* and *oatA*. Only the deletion of *oatA* left the activation σ$^V$ dynamics unchanged (Fig 3E). In the *sigV* mutant, P$_{sigV}$-YFP levels did not increase in response to lysozyme stress (Fig 3C and Appendix Fig S17). Deleting *rsiV* caused all cells to have high P$_{sigV}$-YFP expression even before the addition of lysozyme and the addition of lysozyme did not activate

the system any further (Appendix Fig S17). These findings suggest that the heterogeneity in σ$^V$ activation only depends on σ$^V$ and its anti-sigma factor RsiV (Fig 3F). We found that the leakiness of the inducible *sigV* construct (P$_{hyperspank}$-*sigV*, without addition of IPTG) increased the fraction of activated cells on the addition of lysozyme, as well as causing an increase in the steady-state levels of P$_{sigV}$-YFP before and after stress (Appendix Figs S18 and S19). The leakiness of the RsiV construct (P$_{hyperspank}$-*rsiV*) caused the opposite effect. These effects were still apparent for inducible constructs with lower leakiness (P$_{spank}$-*sigV*, P$_{spank}$-*rsiV* (Appendix Figs S18 and S19)), confirming that *sigV* activation dynamics are sensitive to these two system components.

### Mathematical modelling reveals that P$_{sigV}$-YFP expression dynamics can be explained by the 'mixed' σ$^V$ autoregulatory feedback loop

Based on our transcriptome and overexpression analysis, we constructed a model of a simplified *sigV* operon consisting of *sigV* and its anti-sigma factor *rsiV*, with positive autoregulation of the operon by active σ$^V$ (Fig 3F). We did not include *oatA*, as it is not required for the heterogeneous activation of σ$^V$ (Fig EV3). RsiV binds σ$^V$ to form an inactive complex σ$^V$-RsiV. On stress application, RsiV is degraded and σ$^V$ is free to activate the operon. The model was simulated using Gillespie-type stochastic simulations, tracking the changes in copy number of each species of the system by simulating the individual reaction events (Gillespie, 1977). With the exception of the production of σ$^V$ and RsiV (which were implemented through a Hill function of σ$^V$ activity with a Hill coefficient of 2), all reactions were modelled using mass action kinetics (See Materials and Methods). A Hill coefficient > 1 was required to generate heterogeneous activation dynamics as a degree of

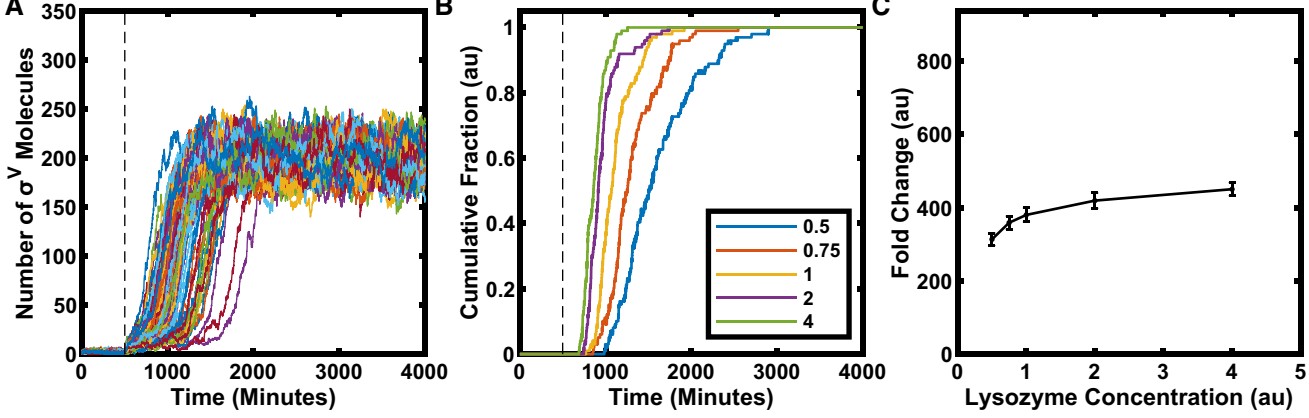

**Figure 4. A model of the simplified σ$^V$ circuit captures experimental results.**

A   Simulations display heterogeneous σ$^V$ activation after lysozyme stress (N = 100). Dashed vertical line marks point of stress addition (parameter *L* changed from 0 to 1) in simulation.

B   In simulations, cells activate σ$^V$ faster as stress levels (the value of the *L* parameter) are increased, as observed experimentally (N = 100 simulations for each stress level: 0.5, 0.75, 1, 2, 4).

C   With increasing stress levels, the steady-state level of σ$^V$ activity increases, as observed experimentally. Bars correspond to the mean ± s.d. (N = 100 simulations for each lysozyme stress level).

Data information: For more information on the number of repeats, please see Appendix Table S3.

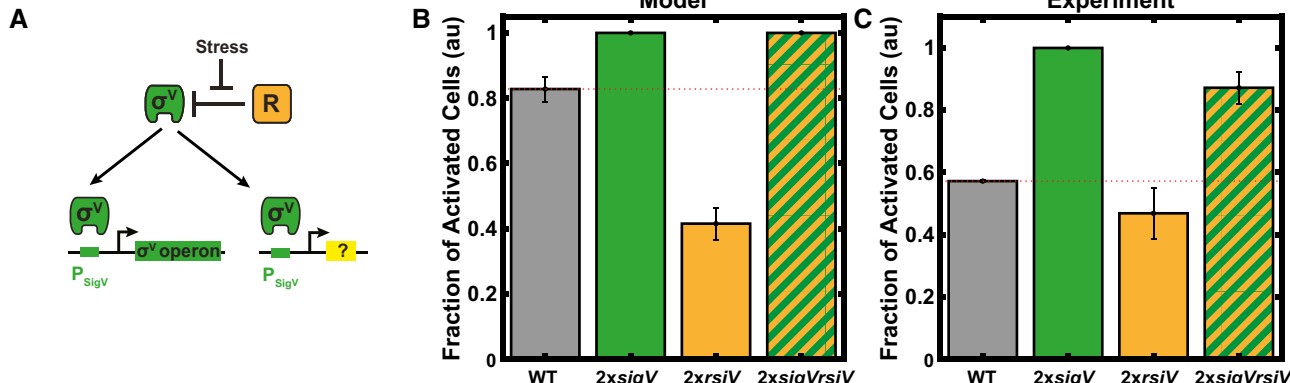

**Figure 5. Phenotypic variability is tuned by additional copies of components of the σ$^V$ circuit.**

A Schematic of genetic perturbations to σ$^V$ circuit. Second copies of σ$^V$ components were added under the control of the P$_{sigV}$ promoter. ? = P$_{sigV}$-sigV (2x*sigV*), P$_{sigV}$-sigVrsiV (2x*sigV-rsiV*) or P$_{sigV}$-rsiV (2x*rsiV*).

B Model simulations predict that the 2x*sigV* or 2x*sigV-rsiV* strains have a homogenous σ$^V$ response to lysozyme, while the 2x*rsiV* strain has increased heterogeneity. Bars correspond to mean ($N$ = 999 simulations) ± s.d. (1,000 bootstraps).

C As predicted, the 2x*sigV* or 2x*sigV-rsiV* strains have a homogenous σ$^V$ response to 1 μg/ml lysozyme, while the 2x*rsiV* strain has increased heterogeneity. Each bar plot is the average of three biological repeats. The bars correspond to the mean ± s.d.

Data information: For both (B) and (C), bars represent the fraction of cells that have activated σ$^V$ 30 min after treatment with lysozyme. σ$^V$ activated cells were defined as those cells that had passed a threshold of the WT mean before lysozyme application plus six standard deviations. For more information on the number of repeats, please see Appendix Table S3.

ultrasensitivity is needed in the system to amplify the response to molecule fluctuations. We note that other networks, including a single positive feedback loop, can generate heterogeneous activation dynamics (See Materials and Methods and Appendix Fig S20), but that this is the simplest model that can simulate the role of both σ$^V$ and RsiV in the regulation of *sigV* activation.

We searched a range of biologically feasible parameters and were able to find parameters that capture the heterogeneous σ$^V$ activation in response to lysozyme stress (Figs 4A and EV4). The *sigV* operon components were assumed to be stable, so the dilution rate was set to approximately match the division rate observed in experiments. Simulations yielded plausible copy numbers for the number of sigma factor molecules (Jishage & Ishihama, 1995; Jishage *et al*, 1996). In addition, we verified that the heterogeneous activation behaviour is robust to perturbations to the system parameters (Appendix Fig S21). Using our model, we were able to recreate several aspects of the experimental data, including the dependency of induction time (Fig 4B) and steady-state levels of *sigV* expression on levels of lysozyme stress (Fig 4C). Finally, the heterogeneous activation dynamics were also modulated by small increases in the baseline production rate of either σ$^V$ or RsiV (Appendix Fig S22), qualitatively matching the effects of the leakiness of the inducible σ$^V$ or RsiV construct observed in experiment (Appendix Fig S18).

Our model consists of a mixed positive and negative feedback loop. We tested the requirements of this feedback for the dynamics by modelling a feedback-broken system, with constitutive expression of *sigV* and *rsiV*. For a range of constitutive expression, the dynamic range of P$_{sigV}$-YFP expression for the feedback-broken system on addition of lysozyme was less than that of the WT system (Fig EV5A and Appendix Fig S23). This reflected the role of the feedback loop in amplifying the system dynamics. To test this prediction

experimentally, we constructed a strain with no autoregulation of the *sigV* operon by knocking out the *sigV* operon and replacing it with a *sigV* operon driven by an inducible promoter. This allowed us to study the system at different steady-state expression levels (by varying IPTG induction level) to the WT system. We found that the fold change induction of the WT on addition of lysozyme was at least 4.5 times higher than that observed in the inducible operon strain, regardless of the IPTG induction level (Fig EV5B and Appendix Fig S24), matching the behaviour observed in our model (Fig EV5A).

### Phenotypic variability is tuned by doubling copy numbers of *sigV* operon genes

To further test the assumptions of our model, we predicted the effect on σ$^V$ activation dynamics of introducing a second copy of each component of the *sigV* operon (*sigV, rsiV* or both *sigV* and *rsiV*), with each component driven by the *sigV* promoter (Fig 5A). Our model predicted that these perturbations would modulate the heterogeneity of the system's response to lysozyme stress. A second copy of *rsiV* meant that large fluctuations in *sigV* expression were required to kick the system into the high σ$^V$ expression state (Fig 5B and Appendix Fig S25). This increased the cell-to-cell variability in response times to lysozyme stress, as compared to the WT in the simulations (Appendix Fig S25). Conversely, with a second copy of *sigV*, or a second copy of both *sigV* and *rsiV,* smaller fluctuations in *sigV* expression are required to kick the system into the high σ$^V$ expression state. Cell-to-cell variability in response times to lysozyme therefore decreases in these simulations (Fig 5 and Appendix Fig S25). In fact, a second copy of *sigV* caused an increase in the levels of σ$^V$ expression even before the addition of lysozyme in the simulations (Appendix Fig S25).

To test these predictions, we constructed strains containing a second copy of either *sigV*, *rsiV* or *sigVrsiV* driven by the *sigV* promoter. As predicted by our model, snapshots of *sigV* expression after induction by 1 μg/ml of lysozyme revealed that a second copy of *sigV* or *sigVrsiV* reduced the observed heterogeneity in σ$^V$ activation, while a second copy of r*siV* caused an increase in the heterogeneity in σ$^V$ activation (Fig 5C and Appendix Fig S26). Finally, adding a second copy of *sigV* alone caused the activation of σ$^V$ even in the absence of stress, as predicted. These results validate our model assumptions and also show that the heterogeneity in *sigV* activation is easily tuned by simple genetic perturbations.

### The heterogeneous activation dynamics of σ$^V$ are dependent on environmental history

Given that the σ$^V$ activation dynamics appear sensitive to the baseline levels of σ$^V$, RsiV and the σ$^V$-RsiV complex, any perturbations to these levels should affect σ$^V$ activation times. We hypothesized that cells should have elevated levels of *sigV* operon components after a lysozyme stress is removed due to the recent high expression of the operon components. The elevated levels of σ$^V$ stored in stabilized σ$^V$-RsiV complexes should cause all cells to respond immediately to a subsequent reapplication of lysozyme. We investigated this hypothesis in our model. We simulated the application of lysozyme stress, which was then removed for the equivalent duration of several cell cycles. The same level of stress was then reapplied and the heterogeneity in σ$^V$ activation disappeared (Fig 6A and Appendix Fig S27). However, as the delay before the second application of lysozyme was increased (allowing system components to relax to pre-stress levels), the heterogeneity gradually returned (Fig 6B and C and Appendix Fig S27). Thus, our simulations predict the existence of a temporary molecular memory of the environmental history.

To test whether the σ$^V$ circuit exhibits such a memory, we grew cells in the mother machine device under 1 μg/ml lysozyme stress before removing the stress for 6 h (approximately seven cell cycles). All cells stopped activating σ$^V$ as soon as lysozyme was removed, as indicated by the decay in YFP fluorescence (Fig 6D and Movie EV2). The decay in fluorescence was tightly synchronized across the population and YFP decayed with a half-life time of 51 ± 1 min, which was similar to the cell cycle time of 51 ± 13 min. When reapplying a 1 μg/ml lysozyme stress after 6 h (approximately seven generations), all cells responded instantaneously. As predicted by the model, the heterogeneity in σ$^V$ activation was lost (Fig 6D, Appendix Fig S28A and Movie EV2). Also as predicted by the model, increasing the duration of the break in stress to 12 h (approximately 14 generations) resulted in heterogeneous activation dynamics similar to those observed in response to the first application of lysozyme stress (Fig 6E and F, Appendix Fig S28B and C and Movie EV3). This return to a heterogeneous response reflects the system's loss of transcriptional memory through reduction in the *sigV* operon component concentrations to pre-stress exposure levels. These results held for two different versions of the mother machine microfluidic device (Appendix Fig S29). Taken together, our results show how the autoregulatory σ$^V$ circuit can generate heterogeneous activation dynamics that can be tuned by stress level, genetic perturbations and the environmental history of the cell.

## Discussion

Here, we report a general mechanism for a bacterial population to tune its phenotypic variability based on stress levels, genetic architecture and environmental history. Using quantitative single-cell microscopy and microfluidics, we found that the activation of the alternative sigma factor σ$^V$ in response to lysozyme stress was heterogeneous. While some cells activated their σ$^V$ pathway immediately, others could take up to six generations to activate their σ$^V$ pathway. The observed phenotypic variability plays a functional role, as cells that respond to a sub-lethal stress were more likely to survive a subsequent higher stress application (Fig 2). Through experiments and modelling, we found that this heterogeneity could be understood by the 'mixed' positive and negative feedback of σ$^V$ activating both itself and its anti-sigma factor RsiV. Alternative sigma factors are a common regulatory system in prokaryotes, often controlling stress response and virulence pathways. The 'mixed' feedback loop is also a common motif, suggesting that this motif can be a general mechanism for bacterial populations to tune their phenotypic variability.

Our modelling and experiments found that recent exposure to lysozyme stress modulates the heterogeneity observed on a second stress application, even though the system turns off after removal of the first lysozyme stress. The key to this behaviour appears to be the 'mixed' feedback loop, as it allows amplified levels of inactive σ$^V$-RsiV complexes in each cell after stress. These complexes can be cleaved by a subsequent addition of lysozyme, releasing σ$^V$ to activate its operon. Similar transcriptional memories of previous stress have been observed in bacterial systems, although typically not to modulate phenotypic heterogeneity. For example, other pathways such as the heat stress response in *B. subtilis* (Runde *et al*, 2014) or the oxidative stress response in yeast (Kelley & Ideker, 2009) have been shown to have a transcriptional memory of past conditions. Often this transcriptional memory is facilitated through an autoregulatory positive feedback loop that can lock the cell into an ON state that is heritable through cell divisions (Novick & Weiner, 1957; Biggar & Crabtree, 2001; Xiong & Ferrell, 2003; Acar *et al*, 2005; Hashimoto *et al*, 2013; Lambert & Kussell, 2014). However, it is difficult for a single positive feedback loop to allow the system to be OFF but primed for future stress, as we find to be the case for the 'mixed' feedback loop in the σ$^V$ pathway. Dilution during growth causes heterogeneous activation of *sigV* to eventually return when levels of σ$^V$-RsiV drop to pre-stress levels, so the memory is qualitatively different from that generated by a positive feedback loop locking a system ON. However, we find that the *sigV* transcriptional memory remains for several generations. In the future, it will be important to investigate whether the 'mixed' feedback loop also tunes the levels of phenotypic diversity by environmental history in other systems. Interestingly, computational studies have proposed that a 'mixed' feedback loop structure in the MAR operon in *Escherichia coli* allows the tuning of the fraction of cells prepared to survive antibiotic exposure (Garcia-Bernardo & Dunlop, 2013). Additionally, the mixed feedback loop mechanism could be compared to other mechanisms proposed to allow the modulation of phenotypic variability, such as multi-site phosphorylation (Libby *et al*, 2019) or threshold-based mechanisms in toxin–antitoxin modules (Rotem *et al*, 2010).

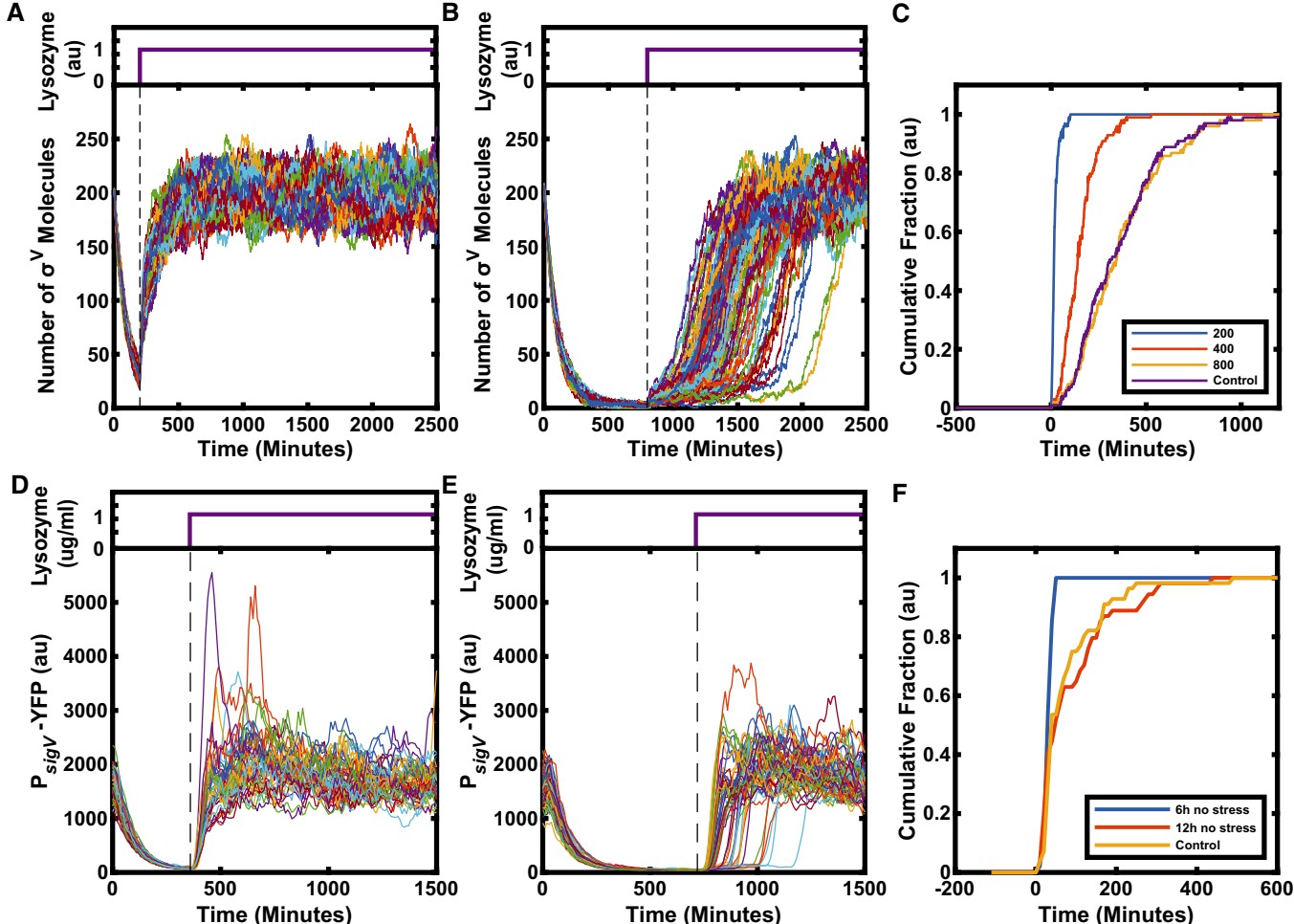

**Figure 6. The σ^V circuit has a memory of previous stress.**

Stress is removed from an activated system at time 0 and reapplied at a time indicated by the dashed line, lysozyme concentrations are indicated by the purple traces above time course plots.

A       Simulations predict that after a break in stress, on reapplication of stress all cells turn on σ^V immediately and homogeneously (N = 99).

B, C    With increasing intervals between stresses, response heterogeneity reappears in the model, indicating a loss of memory. (C) Each line represents the cumulative fraction of cells (N = 99) with σ^V concentration values that are higher than the half maximum of their final values (representing cells that have activated). If the interval between stress is increased long enough, (800 au), the heterogeneity is the same as if the cells had not experienced a prior lysozyme stress (control).

D       Experiments confirm memory in the σ^V circuit. For short intervals between stress applications, cells respond immediately and homogeneously. Each line corresponds to one of 48 single-cell traces.

E, F    For very long intervals between stress (12 h), the heterogeneity is the same as if the cells had not experienced a prior lysozyme stress (control). N > 50 single-cell traces.

Data information: For more information on the number of repeats, please see Appendix Table S3.

While simple, our model allows qualitative matches to data. In future, it will be important to increase the complexity of the model to make more precise predictions of the behaviour of the *sigV* system. One aspect of the system that can be modelled in more detail is how noise in gene expression is generated in the circuit. In our model, we do not model transcription and translation separately and assume uncorrelated noise for each reaction channel. A more detailed model could involve characterizing the noise in terms of bursts of transcription and translation (Friedman *et al*, 2006). In turn, this would require experiments to characterize the noise in transcription, such as single-molecule FISH (Raj &

Oudenaarden, 2008). Our assumption of uncorrelated noise is also a simplification as, for example, we have modelled the degradation events as uncorrelated, which may not hold as these are primarily caused by dilution. Additionally, the system requires ultrasensitivity (in the form of a Hill coefficient $n > 1$ in the operon production term) in order to amplify molecule fluctuations. While other sigma factor response systems have been shown to utilize ultrasensitivity (Narula *et al*, 2012; Narula *et al*, 2016), there is no known source of it in the σ^V circuit (neither the binding of σ^V to RsiV, nor its operon, is cooperative). Further research should examine possible sources of ultrasensitivity in the circuit,

one possibility being sigma factor competition for RNA polymerase (Park *et al*, 2018).

Cells that activate *sigV* quickly after a priming stress have a higher chance of surviving a subsequent high stress (Fig 2B). This points to a potential benefit of early activation of *sigV*. Future work should examine the costs of early *sigV* activation to see if the heterogeneous activation dynamics we observe represent a bet-hedging strategy (Veening *et al*, 2008a; Veening *et al*, 2008b), where early responding cells suffer a fitness penalty in return for protection against future stress. We do not observe a growth rate difference in cells that activate *sigV* earlier compared to later, suggesting that early responders are not suffering a growth rate penalty. However, it is possible that we are missing small growth rate effects, as the time resolution of our mother machine experiments only allows approximately 5 time points to be measured per cell cycle. Our experiments do indicate that high constitutive expression of *sigV* or *oatA* reduces the growth rate in bulk culture (Appendix Fig S30). It could also be that the fitness penalty is due to early *sigV* activation blocking cells from responding to stress with other alternative sigma factors, as it appears alternative sigma factors compete for RNA polymerase (Nyström, 2004; Park *et al*, 2018). Evolution experiments under changeable stressful environments could reveal whether the heterogeneity in activation and transcriptional memory that we observe evolve to optimally match the external environment.

We have found that a combination of noise in gene expression and a mixed feedback loop can generate tunable phenotypic diversity in an alternative sigma factor circuit. Our work will be of utility for synthetic biology, where alternative sigma factors are a promising system for engineering orthogonal gene circuits (Rhodius *et al*, 2013; Bervoets *et al*, 2018; Pinto *et al*, 2018). Going forward, it will be important to observe whether alternative sigma factors more generally are used as a mechanism for bacterial populations to tune phenotypic diversity. This is particularly the case given that alternative sigma factors often control pathways critical to pathogenicity and resistance to antibiotics. Noise in sigma factor activity has already been observed in multiple alternative sigma factor circuits in *B. subtilis* (Locke *et al*, 2011; Narula *et al*, 2016; Park *et al*, 2018), as well as for the general stress response sigma factor in *E. coli* (Patange *et al*, 2018). Additionally, an alternative sigma factor plays a role in generating phenotypic diversity in Mycobacteria (Sureka *et al*, 2008; Ghosh *et al*, 2011). Two other obvious candidates for further study are the pathogens *Enterococcus faecalis* and *Clostridioides difficile* (Ho & Ellermeier, 2019), which also have a σ$^V$ pathway that is responsive to lysozyme.

# Materials and Methods

### Strains and media

All strains are derivatives of the PY79 background strain (Appendix Table S1). Deletions were generated by replacing genes of interest with an antibiotic resistance cassette by recombination of a linear DNA fragment homologous to the region of interest. All strains had a Δ*ytvA*::neo deletion insertion. YtvA is a blue light sensor, and it was deleted to avoid any activation of the general stress response pathway σ$^B$ by the microscope illumination (Gaidenko *et al*, 2006; Locke *et al*, 2011). For strains used in mother machine experiments, cells were made immotile by inserting a Δ*haG*:erm deletion cassette in order to improve the loading into the microfluidic device. Additionally, all strains contained a house keeping σ$^A$ promoter-driven mCherry for segmentation and as a constitutive control (Locke *et al*, 2011).

Cells were routinely grown in Spizizen's Minimal Media (SMM) (Spizizen, 1958). It contained 50 μg/ml tryptophan as an amino acid source and 0.5% glucose as a carbon source. Cultures were started from frozen stock in SMM and grown overnight at 30°C to an OD between 0.3 and 0.8. The overnight cultures were resuspended to an OD of 0.01 and regrown to an OD of 0.1 at 37°C.

### Plasmids

*Escherichia coli* strain DH5α was used to clone all plasmids. The cloning was done with a combination of non-ligase-dependent cloning and standard molecular cloning techniques using Clontech In-Fusion Advantage PCR Cloning kits. Plasmids were chromosomally integrated into the PY79 background via double crossover using standard techniques. The list below provides a description of the used plasmids, with details on selection marker and integration position/cassette given at the beginning. Note that all plasmids below replicate in *E. coli* but not in *B. subtilis*.

1   *ppsB*::P$_{trpE}$ -mCherry Phleo$^R$
    This plasmid was used to provide uniform expression of mCherry from a σ$^A$-dependent promoter, enabling automatic image segmentation (cell identification) in time-lapse movie analysis. A minimal σ$^A$ promoter from the *trpE* gene was cloned into a vector with ppsB homology regions (Locke *et al*, 2011). The original integration vector was a gift from A. Eldar (Eldar *et al*, 2009).

2   *sacA*::P$_{sigV}$ -YFP Cm$^R$
    The target promoter of *sigV* was cloned into the EcoRI/BamHI sites of AEC12 (Eldar *et al*, 2009) (gift from M. Elowitz, CalTech).

3   *amyE*::P$_{sigV}$ -mTurq Spect$^R$
    The target promoter of *sigV* and the mTurq gene from GL-FP-31 (gift from E. Gardner, Harvard) were cloned into the sites EcoRI/BAmHI and BamHI/HindIII of pdL30 (Locke *et al*, 2011).

4   *amyE*::P$_{hyperspank}$ -X Spect$^R$
    Where X is *sigV*, *rsiV*, *yrhK*, *oatA*, *rasP*, *sipS*, *pbpX* or YFP. The coding region of the genes along with a 5′ transcriptional terminator was cloned downstream of the hyperspank IPTG-inducible promoter in plasmid pDR-111 (gift from D. Rudner, Harvard).

5   *amyE*::P$_{spank}$ -X Spect$^R$
    Where X is *sigV*, *rsiV*, *yrhK*, *oatA*, *rasP*, *sipS*, *pbpX* or YFP. The coding region of the genes along with a 5′ transcriptional terminator was cloned downstream of the spank IPTG-inducible promoter in plasmid pDR-110 (gift from D. Rudner, Harvard).

6   *amyE*::P$_{sigV}$-X Spect$^R$
    The coding region of (where X is *sigV*, *rsiV* or *sigVrsiV*) along with a 5′ transcriptional terminator was cloned downstream of the *sigV* promoter.

## Microscopy

A Nikon inverted Ti-E microscope (Nikon, Amsterdam, Netherlands) with a Nikon Perfect Focus System (PFS) hardware autofocus was used. All images were acquired using either a Photometrics CoolSnap HQ2 CCD camera (Photometrics, Tucson, AZ, USA) or a Photometrics Prime sCMOS camera (Photometrics, Tucson, AZ, USA). The microscope stage was enclosed in an incubator (Solent Scientific, Segensworth, UK) which was set to 37°C for all experiments. Illumination was provided by a LED lamp (CoolLED, Andover, UK). Epifluorescence was provided by a Lumencore Solar II light engine (Lumencore, Beaverton, OR, USA). Chroma filters (Chroma, Bellows Falls, USA) #41027 for the RFP channel, #49003 for the YFP channel and #49001 for the CFP channel were used. All experiments were done with a phase 100x Plan Apo (NA 1.4) objective (Nikon, Amsterdam, Netherlands). Metamorph (Molecular Device, Sunnyvale, CA, USA) controlled the camera, the motorized stage (Nikon, Amsterdam, Netherlands) and the microscope.

### Snapshots

Agarose pads were made by pipetting 1 ml of 1.5% (wt/vol) low melt agarose (Merck, Darmstadt, Germany) dissolved in PBS onto a 22 mm$^2$ cover glass slide. Immediately after pipetting, another cover glass slide was placed on top of the agarose creating a "sandwich" with the agarose in the middle. Once the agarose had solidified, the top cover glass was removed and the agarose was cut into squares of approximately 5 mm × 5 mm with a scalpel.

Once the cultures had reached an OD of 0.1, they were split into aliquots of 1 ml to which different concentrations of lysozyme from hen eggs white (Sigma Aldrich, St. Louis, MO, USA) were added. The aliquots were then incubated at 37°C for 30 min. For experiments with IPTG-inducible strains, 1 mM of IPTG was added to the aliquots for 60 min before addition of lysozyme, in order to allow for full induction of the promoter before stress. Different concentrations of lysozyme were then applied for 30 min before snapshot measurements. For snapshots, 2 μl of cell culture was pipetted onto the pads, which were left to dry and then laid face down onto a cover glass-bottom dish (#HBSt-5040, WillCO-dish, Amsterdam, Netherlands). The glass dish was sealed with parafilm and put under the microscope to image. Single-cell data were extracted using custom MATLAB (Mathworks, Natick, USA) scripts based on the Schnitzcells package (Young *et al*, 2011).

### CellAsic experiments

Overnight cultures were grown and then resuspended to an OD of 0.01 as described above. Once the cultures had reached an OD of 0.1, they were resuspended to an OD of 0.001 for loading into CellAsic B04 microfluidic chips (Merck, Darmstadt, Germany). Cells were loaded into the microfluidic chips with a pressure of 4–6.5 psi for 2–4 s. The following lysozyme concentrations were investigated with the CellAsic setup: 0, 0.5, 1 and 4 μg/ml. Fresh media was perfused into the chip with a pressure of 1 psi. After 60 min of growth in standard SMM, the media was switched to media containing lysozyme. Cells were imaged at regular intervals (every 10 min), and the acquired movies were analysed with the standard Schnitzcells package for MATLAB (Young *et al*, 2011).

## Microfluidics

### Wafer fabrication

We used two different microfluidic designs of the mother machine:

1   All the mother machine data in this paper were acquired with a microfluidic design based on the original mother machine (Wang *et al*, 2010) unless stated otherwise. As a substrate for the PDMS chip fabrication, we used an epoxy master which was a kind gift from the Jun laboratory at the University of California, San Diego, USA (Taheri-Araghi *et al*, 2015).

2   For data shown in Appendix Fig S29, a different mother machine design was used. It was based on a mother machine design first described by Norman *et al* (2013). The main difference to the Jun laboratory design is that the channels in which cells grew were longer and that they have a shallow side channel for better perfusion of media to the cells. As a substrate for the PDMS chip fabrication, we used an epoxy master which was a kind gift from the Elowitz laboratory at the California Institute of Technology, Pasadena, USA (Park *et al*, 2018).

### Soft lithography

Sylgard 184 polydimethylsiloxane (PDMS) from Dow Corning (Midland, MI, USA) was used to replica-mould the microfluidic devices. A 10:1 (base:curing agent) PDMS mixture was cast onto the epoxy master and cured overnight at 65°C. In the next morning, the chips were cut out and the inlets were punched using a 0.75 mm biopsy puncher (#504529, WPI, Sarasota, FL, USA). To remove any uncured PDMS from the chips, these were chemically treated in a pentane (Sigma Aldrich, St. Louis, Mo, USA) bath for 90 min and then washed twice for 90 min in acetone (Sigma Aldrich, St. Louis, Mo, USA). The chips were then left to dry overnight in the fume hood and were ready to use the next day (Gruenberger *et al*, 2013).

### Experimental preparation

The chips were plasma bonded to a glass-bottom dish (HBSt-5040, Willco Wells, Amsterdam Netherlands) by treating the chip and the glass dish surfaces with a plasma (Femto Plasma System, Diener, Ebhausen, Germany) for 12 s at a power of 30 W and a pressure of 0.35 mbar. The chips were then put into the oven for 10 min at 65°C, in order to strengthen the bonding. In the meantime, a bovine serum albumin (BSA) passivation buffer was prepared at a concentration of 20 mg/ml in SMM. The chips were then passivated with this buffer for 1 h at 37°C.

Cells were grown overnight as for the other movie experiments. 10 ml of the regrown cultures with an OD of 0.1 was spun down for 10 min at 3,000 *g* (4,000 rpm) and resuspended in 0.1 ml to a final OD of 10. This cell suspension was loaded into the growth channels by spinning the chip for 5 min at 3,000 rpm with a spin coater (Polos SPIN150i, SPS, Harlem, the Netherland).

### Setup of microfluidic mother machine experiments

Perfusion was provided by a Fluigent pressure pump setup. Briefly, reservoirs were pressurized using a computer controllable pressure controller (MFCS-EZ, 1 bar, Villejuif, Fluigent, France). A rotary selection valve (M-Switch, Fluigent, Villejuif, France) allowed for switching between different reservoirs. The flow rate was set to

1 ml/h and by using a flow sensor (M-Flow Sensor, Fluigent, Ville-juif, France) which feeds back onto the pressure controller.

For the experiments on the memory in the $\sigma^V$ response, syringe pumps (Legato 200, KD Scientific, Holliston, USA) were used. Switching was provided by an electronic switching valve (MXP9960, Rheodyne, Lake Forest, USA). The flow rate remained at 1 ml/h.

The analysis of all movies was done using a modified version of Schnitzcells (Young *et al*, 2011; Park *et al*, 2018) optimized for mother machine experiments.

### Memory experiments

For all shown memory experiments, the JLB221 strain (ΔespH) was used instead of JLB130, unless stated otherwise. This was done in order to prevent clogging of the microfluidic chip during these long-term experiments due to the production of extracellular matrix. The two strains had the same $P_{sigV}$-YFP dynamics in response to lyso-zyme (Appendix Figs S31 and S32).

Cells were inoculated in SMM and grown overnight to an OD of 0.3–0.8. In the morning, the cells were resuspended to an OD of 0.01 in SMM + 1 μg/ml lysozyme and grown to an OD of 0.1. Cells were then loaded as described in the "Experimental preparation" section.

In the mother machine, the cells were grown for 4 h with 1 μg/ml lysozyme before removing the stress. The stress was then reapplied after 6 or 12 h.

### Growth curve determination

Cultures were started from frozen stock in SMM and grown overnight at 30°C. The overnight cultures were resuspended to an OD of 0.02 and grown for 2 h at 37°C before adding 1 mM of IPTG. The cells were then grown for another hour before adjusting the OD to 0.1 and aliquoting the culture into 2 ml samples (with one 2 ml sample used for each time point), and then, the samples were returned to grow at 37°C. A sample was taken every hour, and the OD600 was determined.

### Data analysis

### Cumulative activation time

The activation time in mother machine experiments was calculated as the time between switching from SMM to lysozyme and the time point when then $P_{sigV}$-YFP passed its half maximum (Appendix Fig S1). The cumulative fraction was then calculated as the cumulative fraction of cells with $P_{sigV}$-YFP values higher than the half maximum of their final values (representing cells that have activated) (Appendix Figs S1 and S8).

### Removal of overshooting cells

For 4 μg/ml lysozyme, the $P_{sigV}$-YFP activity overshot its steady-state activity. Overshooting cells were sick and also wider. Thus, overshooting cells could be removed based on their width. First, the single-cell width traces were smoothed with a Gaussian filter, and their maximum value was determined. All single-cell traces with a maximum width larger than the mean width + 6 sigma of cells before the stress were removed (Appendix Figs S9 and S10).

### Growth rate

The instantaneous growth rate as shown in Appendix Fig S11 was calculated as in (Martins *et al*, 2018):

$$Gr = \frac{\log(len_{i+1}) - \log(len_i)}{\Delta t} \tag{1}$$

where $len_i$ is the length of the cell at frame $i$, and $\Delta t$ is the time between subsequent frames taken from the image metadata. Growth rates were only calculated within one cell cycle and not over a division.

### Calculation of surviving cells in priming experiment

The lysis curve (Fig 2B) was calculated as the fraction of surviving cells 280 min after the addition of 20 μg/ml lysozyme. We only examined the survival rate of the top three cells in each channel at the time of the addition of 20 μg/ml lysozyme to avoid problems with cells being washed out of the chip during the 280 min after the stress. Only one surviving lineage (the longest) was counted from each of these top three cells, to avoid cell division artificially increasing the number of survivors. The survival rate at the end of the movie using this method was 12.5 ± 2.7% (Fig 2B). To verify this method, we also used an alternative approach. In this second method, the end of each channel was approximated using the average cell position before a cell left the channel in the ~ 100 frames before the stress was added. Cells that at their last recorded position after the addition of 20 μg/ml lysozyme stress were within six standard deviations of the end of the channel were removed from the analysis as they were assumed to have left the channel rather than died. This second approach gave qualitatively similar results, with a survival rate of 13.6 ± 3.1%.

### Statistical analysis of the effects of higher $P_{sigV}$-YFP levels before lysozyme stress application on response time

As the distributions in Appendix Fig S14 were not normal, we use the non-parametric Kolmogorov–Smirnov test (ks test) to verify the null hypothesis that cells with higher $P_{sigV}$-YFP values before lysozyme stress application would have the same activation time distribution as cells with lower $P_{sigV}$-YFP expression. Cells with high $P_{sigV}$-YFP were defined as cells with a YFP fluorescence higher than the mean YFP fluorescence before the addition of lysozyme plus one standard deviation. All other cells were defined as low $P_{sigV}$-YFP. A *P*-value below 0.05 was used to reject the null hypothesis. The ks test was performed using the built in MATLAB function *kstest2*.

We found that for 0.5, 1 and 2 μg/ml lysozyme the null hypothesis was rejected. Thus, cells with higher $P_{sigV}$-YFP values before lysozyme stress application had a shorter activation time. For 4 μg/ml lysozyme, the null hypothesis was accepted. Therefore, cells with higher $P_{sigV}$-YFP values before lysozyme stress application had the same activation time as cells with lower $P_{sigV}$-YFP values.

### Calculation of fraction of activated cells from snapshots

Cells which, after the application of lysozyme stress, had a higher mean $P_{sigV}$-YFP expression than the mean $P_{sigV}$-YFP expression plus six standard deviations before stress were defined as having activated. The fraction of activated cells was then calculated as the number of cells which had activated normalized by the total number of cells (Figs 3 and 5).

## RNA-seq experiment

### Lysozyme treatment before RNA extraction

Overnight cultures were prepared as for snapshot or movie experiments. Briefly, cultures were started from frozen stock in SMM and grown overnight at 30°C to an OD between 0.3 and 0.8. The overnight cultures were resuspended to an OD of 0.01 and regrown to an OD of 0.1 at 37°C. Once the cultures had reached an OD of 0.1, they were split into aliquots of 10 ml to which either 0 or 1 µg/ml lysozyme from hen eggs white (Sigma Aldrich, St. Louis, MO, USA) were added. The aliquots were then incubated at 37°C for 30 min.

### RNA extraction and library preparation

10 ml of *B. subtilis* at OD 0.1–0.2 was centrifuged in 15-ml falcon tubes for 10 min at 3,000 *g* (4,000 rpm). Cell pellets were resuspended in 1 ml of Qiagen RNAprotect Bacteria Reagent and flash frozen in liquid nitrogen.

Defrosted cells were pelleted by a quick centrifugation. After removing the supernatant, cells were resuspended in 1 ml buffer RLT from Qiagen RNeasy kit. Resuspended cells were transferred in a Fastprep Lysing Matrix B tube (MP Bio) and processed in Fastprep apparatus 45 s at speed 6.5 M/s. 700 µl of the supernatant, containing lysed cells, was transferred to a new microcentrifuge tube, to which 500 µl of absolute ethanol was added. After vortexing, the lysate was transferred to a RNeasy spin column and centrifuged 15 s at > 9,400 *g* (10,000 rpm). RNA purification was then carried out following the instructions of the Qiagen RNeasy kit. RNA quality and integrity were assessed on the Agilent 2200 TapeStation, and RNA concentration was assessed using Qubit RNA HS assay kit. Library preparation was performed using ScriptSeq™ Complete Kit (Illumina, BB1224), for 2 µg of high integrity total RNA (RIN > 8). The libraries were sequenced on a NextSeq500 using paired-end sequencing of 75 bp in length.

### RNA-seq analysis

The raw reads were analysed using a combination of publicly available software and in-house scripts. We first assessed the quality of reads using FastQC (www.bioinformatics.babraham.ac.uk/projects/fastqc/). Reads were aligned to the *B. subtilis* PY79 transcriptome (NCIB no.CP006881) using Salmon (Patro *et al*, 2017). Read counts for each gene were imported using the tximport R package (Soneson *et al*, 2015). Genes differentially expressed (*P*-value < 5%) between the WT and *ΔsigV* mutant, or in response to lysozyme treatment were identified using the DESeq2 R package (Love *et al*, 2014). The RNA-seq data of our study can be found here (Gene Expression Omnibus: GSE171761, https://www.ncbi.nlm.nih.gov/geo/query/acc.cgi?acc = GSE171761).

## Mathematical model

We constructed a mathematical model of the $\sigma^V$ circuit that was based on the core components that we found experimentally to modulate the heterogeneity in $\sigma^V$ activation (Fig 3). The model consists of $\sigma^V$, RsiV and the $\sigma^V$-RsiV complex. Free $\sigma^V$ that is not bound in a complex activates the production of both $\sigma^V$ and RsiV. Lysozyme stress was modelled as activating the cleavage of the $\sigma^V$-RsiV complex, releasing $\sigma^V$ while degrading RsiV. We note that although heterogeneous activation can be simulated through just a positive feedback loop, (Tiwari *et al,* 2011; Frigola *et al,* 2012)

(Appendix Fig S20), our model is the simplest model that could be compared to the mutations including both *sigV* and *rsiV* (Fig 5).

### Model reactions

We used the following set of biochemical reactions to model the dynamics of $\sigma^V$, RsiV and $\sigma^V$-RsiV.

Production:

$$\varnothing \rightarrow \sigma^V + \text{RsiV}$$

The production rate of $\sigma^V$ and RsiV was assumed to follow a Hill function, where the production rate of both $\sigma^V$ and RsiV were identical and activated by $\sigma^V$. The production rate is as follows: $v_0 + v*(\sigma^V)^n/((\sigma^V)^n + K^n)$, where $v_0$ (= 0.1 molecules/min, is the operon leakage), $v$ (= 2.5 molecules/min, is the maximal operon activity), $K$ (= 60 molecules, is the apparent dissociation constant) and $n$ (= 2, is the Hill coefficient) are parameters.

Dilution/Degradation:

$$\sigma^V \rightarrow \varnothing.$$

$$\text{RsiV} \rightarrow \varnothing$$

$$\sigma^V\text{-RsiV} \rightarrow \varnothing$$

We assumed a constant and identical dilution rate for all three components of the system. The dilution rate is set by the parameter $k_{deg}$ (= 0.01 min$^{-1}$).

Binding/Dissociation:

$$\sigma^V + \text{RsiV} \rightarrow \sigma^V\text{-RsiV}$$

$$\sigma^V\text{-RsiV} \rightarrow \sigma^V + \text{RsiV}$$

We assumed that $\sigma^V$ and RsiV would bind to each other to form a complex and that this complex could dissociate. We also assumed the binding rate to be higher than the dissociation rate. The binding rate is set by the parameter $k_B$ (= 10 molecules$^{-1}$ min$^{-1}$) and the dissociation rate by the parameter $k_D$ (=5 min$^{-1}$).

Cleavage:

$$\sigma^V\text{-RsiV} \rightarrow \sigma^V$$

We assumed that the $\sigma^V$/RsiV complex would be cleaved by lysozyme, with $\sigma^V$ getting released and RsiV degraded. The cleavage rate is set by the product of the two parameters $L$ (which attains variable values, is the amount of lysozyme stress experienced by the system) and $k_C$ (= 0.05 min$^{-1}$, is the base cleavage rate of the complex). Splitting the rate into two parameters allows setting more natural lysozyme input values (such as 1 and 2).

### Model implementation

The model was implemented in the Julia programming language using the Catalyst.jl modelling package. Simulations were made using the DifferentialEquations.jl package (Rackauckas & Nie, 2017b). To account for the stochastic nature of the system, we use a Gillespie-type model (Gillespie, 1977). Here, we tracked the copy numbers of the three components of the system (σV, RsiV and σV-RsiV), and their change due to the individual reaction events. We

used Gillespie's direct stochastic simulation algorithm to determine the time to, and which, the next reaction event in the simulation should be. We used DifferentialEquations.jl's SSAStepper method to simulate the model. For more details, please see the implementations in the files provided. The model parameters were hand-picked by carrying out a coarse-grained search of feasible values. The degradation rate was set to be similar to what would be produced by the bacteria division rate in the experiments. This yields plausible copy numbers for the number of sigma factor molecules (Jishage & Ishihama, 1995; Jishage *et al*, 1996). See Appendix Table S2 for parameter values. The heterogeneous activation behaviour is robust with respect to perturbations to the selected parameters (Appendix Fig S21). Finally, we also note that a continuous SDE model, based on the chemical Langevin equations, could reproduce the features of the system, (Appendix Fig S33), suggesting that that behaviour does not depend on choice of model approach (Gillespie, 2000; Rackauckas & Nie, 2017a).

## Data availability

The RNA-seq data produced in this study can be found here:

Gene Expression Omnibus GSE171761: https://www.ncbi.nlm.nih.gov/geo/query/acc.cgi?acc = GSE171761.

Model simulation, data analysis and figure plotting code, as well as source data for main and extended view figures can be found here: https://gitlab.com/slcu/teamJL/schwall_etal_msb_2021.

**Expanded View** for this article is available online.

## Acknowledgements

We thank John Sauls and Professor Suckjoon Jun for the kind gift of a mother machine epoxy mould. We thank Jin Park for help with the mother machine analysis code, Michael Elowitz for sharing plasmids, Craig Ellermeier for sharing strains, critical reading of the manuscript and helpful discussions, Hugo Tavares for help with RNA-seq analysis, Katie Abley for critical reading of the manuscript and Niklas Korsbo for helpful discussions. This research was made possible by the award of a European Research Council under the European Union's Seventh Framework Programme (FP/2007-2013)/ERC Grant Agreement 338060. The work in the Locke laboratory is further supported by a fellowship from the Gatsby Foundation (GAT3272/GLC). Torkel Loman has received funding from the European Union's Horizon 2020 research and innovation programme under the Marie Skłodowska-Curie grant agreement No. 721456.

## Author contributions

CPS, TEL and JCWL conceived and designed the study. CPS, TEL, BMCM and JCWL analysed and interpreted the data and wrote the article. CPS, VK and TL performed the experiments. SC did the RNA-seq experiments. CV and TS constructed strains. TEL and BMCM developed the mathematical model.

## Conflict of interest

The authors declare that they have no conflict of interest.

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
