## [Review Process File · Molecular Systems Biology]

Tunable phenotypic variability through an autoregulatory alternative sigma factor circuit

Christian Schwall, Torkel Loman, Bruno Martins, Sandra Cortijo, Casandra Villava, Vassili Kusmartsev, Toby Livesey, Teresa Saez, James Locke

DOI: [10.15252/msb.20209832](https://doi.org/10.15252/msb.20209832)

Corresponding author(s): James Locke (james.locke@slcu.cam.ac.uk)

Review Timeline:

Submission Date:	3rd Jul 20
Editorial Decision:	28th Aug 20
Revision Received:	18th Apr 21
Editorial Decision:	12th May 21
Revision Received:	28th May 21
Accepted:	1st Jun 21

Editor: Jingyi Hou

Transaction Report:

Thank you for submitting your work to Molecular Systems Biology. We have now heard back from two of the three reviewers who agreed to evaluate your manuscript. Unfortunately, after a series of reminders we did not manage to obtain a report from reviewer #3. In the interest of time, and since the recommendations of the other two reviewers are quite similar, I prefer to make a decision now rather than further delaying the process. As you will see from the reports below, the reviewers acknowledge the potential interest of the study. They raise however a series of concerns, which we would ask you to address in a major revision.

Since the reviewers' recommendations are rather clear, there is no need to reiterate all the points listed below. Most of the reviewers' concerns are related to the mathematical modelling and the interpretation of the experimental data, which need to be addressed carefully.

All other issues raised by the reviewers need to be satisfactorily addressed as well. As you may already know, our editorial policy allows in principle a single round of major revision and it is therefore essential to provide responses to the reviewers' comments that are as complete as possible.

On a more editorial level, we would ask you to address the following issues.

REFEREE REPORTS

Reviewer #1:

In this study, the authors find that the *B. subtilis* lysozyme resistance response is activated non-uniformly across a population, in a way that depends on the stress level and environmental history. Through experiments that use varying lysozyme levels, genetic manipulations, and mathematical modeling, the authors find that this heterogeneity is mediated by a mixed positive/negative feedback network with the alternative sigma factor sigmaV at its core. This study adds to the growing body of evidence showing that gene regulation in bacteria is much more dynamic and heterogeneous than traditionally thought.

Overall, the study is well conducted, and the conclusions are supported by the experimental data. On the other hand, addressing raised technical concerns about the model and some additional experiments could further strengthen the paper conclusions.

Major Points:

-While the model is qualitatively and, in some aspects, quantitatively matches the experimental data, several technical concerns about the model ingredients make me doubt that the heterogeneity mechanism is the model is realistic. Therefore, hypothesized heterogeneity mechanism may not be the right one. Most critically, the authors use Hill equation with hill coefficient $n=4$ to describe the gene activation by σ^V . Given that σ factor acts as monomeric subunit that binds to RNAP core, this cooperativity is not justified. I am certain, that both deterministic bifurcation diagram of the model showing irreversible bistability and heterogeneity of stochastic switching times that are likely based on critical slowdown resulting from this bistability critically depend on this unrealistic assumption. The authors need to verify if the model can match experimentally observed behaviors for $n=1$ and if not look for other explicit mechanism (e.g. sequestration) that are capable of producing ultrasensitive and/or bistable response is the system.

-The formalism the authors have chosen to simulate stochastic dynamics have multiple problems and need to be verified against more basic approaches, e.g. SSA simulations of the elementary reactions. First, Chemical Langevin formalism is developed for a specific types of chemical systems, in which time-steps can be selected such that a large number of reactions occur per unit time and rate functions (propensities) remain approximately constant during the same interval. The system simulated is unlikely to satisfy these assumptions (I may say it almost specifically designed to violate these). It is stiff with fast post-translational interactions (binding of σ /anti- σ) much faster than protein production/degradation. It has two sources of ultrasensitivity (high Hill coefficient noted above) but also large changes in free σ when $\sigma_{total}:\text{anti-}\sigma_{total}$ ratio changes for below to above 1. Second, the authors ignore bursty nature of gene expression that is widely shown to be main contributor to intrinsic noise (need to separately simulate mRNA and Proteins to properly account for translational bursting or use well established phenomenological approximation for proper modeling of bursty expression. Third, uncorrelated noise in reaction channels assumed may not be a valid assumption. For instance, if σ /anti- σ are stable and mainly degraded via dilution, the noise in the corresponding terms must be correlated but simulated independently (N5,N6,N7 terms)

-The main conclusion of the paper relates the phenotypic variability to the autoregulation of operon which includes σ^V , rsiV , oatA . (with the last one being not critical for the mechanism). I am surprised the authors did not attempt to test this claim directly with replacing the autoregulated promoter of the inducible one. With a recent development of CRISPR-assisted dCas9 activation systems in bacteria such as experiment should be technically feasible.

Other comments:

-While it is clear that the σ^V network allows cells to modulate their heterogeneity, there is not discussion about why they would want to do that instead of homogeneously switching the resistance response on. Can the authors speculate on this? Is there any previous evidence on a detrimental effect of expression of any of the σ^V -regulated genes? If not, a potential fitness

cost of prolonged and unnecessary σ^V activation could be revealed by measuring growth from wild-type cells vs. the σ^V+ strain in the absence of lysozyme.

-In Figure S6, heterogeneity in experiments conducted in an alternative microfluidic device is shown. However, this heterogeneity does not seem to decrease with increasing lysozyme concentration. Can the authors comment on this? Perhaps the experiment needed to go for longer.

-The experiment in Figure 2 needs to be described more precisely in the main text. There, it is mentioned that the end of the movie is "280 min after the addition of 20 $\mu\text{g/ml}$ of lysozyme". However, it is not clear if this is the case for both priming and non-priming conditions, and in fact Figure S16 shows that it is not. One could think that this difference in timing could explain the observed difference in survivors. However, Figure S16 also shows that non-primed cells die almost immediately after application of 20 $\mu\text{g/ml}$ lysozyme, which is not mentioned in the main text. The diagram with no numbers in the x axis shown in Figure 2A does not help clarify this. I recommend replacing Figure 2A with the two panels in Figure S16. Furthermore, the fact that non-primed cells die immediately should be explicitly mentioned in the main text and/or the figure legend.

-The plots in Figures 3D and E are a little confusing. First, it is not clear why $\text{P}\sigma^V\text{-YFP}$ expression is being shown, given that we were concerned about heterogeneity in activation time and not necessarily expression. Furthermore, the main text should make it more clear when measurements were taken in relation to lysozyme and IPTG application.

-Could the experiment in Figure 3E be conducted with $\text{rsiV}+$ too? If not, why?

Reviewer #2:

Summary

The authors investigate the stress response of *B. Subtilis* to lysozyme, an enzyme that is produced by the immune system to kill bacteria via cell wall lysis. The genetic components involved in this stress response are well known. The authors leverage this knowledge to investigate how the genetic architecture governs the cell response. They start by measuring the expression levels of a key regulator, σ^V , in response to lysozyme stress. Importantly, going beyond previous work, they measure the single-cell diversity by using state-of-the-art microfluidics (the mother machine). They find a highly heterogeneous stress response, especially at low stress. Further, exposure to low concentrations of lysozyme 'primes' cells by increasing the level of σ^V ; this protects cells from subsequent higher doses of lysozyme. Based on RNA-seq data and a $\Delta\sigma^V$ mutant they suggest that no other pathways are involved in the stress response. Using multiple fluorescent reporter genes, they next dissect the σ^V circuit and conclude that only two components, σ^V and RsiV are responsible for the single-cell heterogeneity. A model of this minimal circuit of two components can indeed reproduce key characteristics of the single-cell heterogeneity (stochastic switching, increased heterogeneity at low stress and a stronger switch at higher stress). Lastly, they show that the σ^V circuit can 'memorize' previous stresses as elevated σ^V levels only gradually decay upon removal of lysozyme stress.

General remarks

The study is well motivated, well described and, technically, seems largely thorough. The topic is relevant for microbiologists interested in lysozyme stress and alternative sigma factors, and more broadly for systems biologists and biophysicists interested in the effects of phenotypic heterogeneity. While this work does not provide new insight into molecular interactions, it provides a

rationale for the architecture of the σV regulatory network. Heterogeneous stress responses via discrete (meta-)stable states have been shown before, but how cells generate and control such heterogeneity using genetic circuits is rarely understood. This work makes progress in this direction. However, I have several concerns regarding the interpretation of the experimental data, the mathematical model, and the design of some of the experiments, which need to be addressed.

Major points

The effects of changes in mean expression level and expression heterogeneity need to be disentangled more clearly. The problem is that any change in mean expression level will generally affect heterogeneity, with higher expression levels leading to lower heterogeneity in relative terms (quantified e.g. by the variation coefficient). This has been established in various genome-wide studies (see e.g. Fig. 1F in PMID 22275871, Fig. 2B in PMID 20671182, Fig. 2a in PMID 16715097). This phenomenon needs to be taken into account when interpreting experiments in which the system is perturbed and the mean expression level and its variability change. This affects the interpretation of several experiments but it is particularly important for conclusions as in l. 7-8 on p. 10: given the studies above, it is clear that heterogeneity in expression can be tuned by genetic changes that affect the mean expression level. It would be a much more convincing point if the authors can show that this tuning can be done without changing mean expression level (or if the effects go far beyond those expected from changes in the mean alone).

It is unclear if overexpression (and deletion) of circuit components (p. 7, l. 24-25) can really disentangle which components contribute to the heterogeneity of the response. These are drastic perturbations, which affect mean expression levels (see previous point). Even if they don't, it is not obvious what we can conclude from an effect of overexpressing component A on the expression variability of component B without also quantifying the variability of component A (and correlating it with that of B). The rationale for this needs to be explained in more detail.

Using this approach, the authors find that the expression level of three genes is affected by lysozyme stress (σV , *RsiV* and *oatA*) but conclude that *oatA* is not involved in the heterogeneity of the stress response, as σV expression is still heterogeneous at elevated levels of lysozyme when *oatA* is overexpressed. Therefore, they conclude that a minimal network to explain heterogeneity consists of σV and *RsiV*. However, no check is performed with an *RsiV*⁺ mutant. To investigate whether *RsiV* is indeed part of the minimal circuit needed to explain heterogeneity, a similar check should be done for this protein as for *oatA* in Fig 3E.

The verification part of the model (page 9) is not very convincing. Basically, the authors show that the mean time and standard deviation scale with each other. However, this is not only true for this specific genetic network, but for just about any model that considers stochastic transition events. The authors should at least mention that these predictions are not particularly unique for this specific model/network motif. To verify the model further, the fit parameters should be explained in more detail. How many free parameters are there? Are they constrained by the data? Are their values plausible based on literature knowledge? Since it is a stochastic model, it should also be possible to comment on the copy numbers of the circuit components in the model, which should be compared to experimental estimates.

This work needs to be better put into the context of the field. For several points, recent single-cell studies on other microbial stress responses made conceptually similar observations. For example, an increase of gene expression variability when an inducer (or stressor) concentration is decreased (p. 6, l. 10-12) is often observed for gene expression responses, see e.g. Fig. 2 in PMID 18469087. A

conceptually similar single-cell experiment about 'priming' bacteria before a higher stress is added was recently done in PMID 28342718 and could be mentioned here.

Minor points

Showing a dose response curve (growth rate vs. [lysozyme]) would be helpful to put the used concentrations into context.

The authors show that high levels of σ^V protect cells against lysozyme exposure, and that a heterogeneous cell response therefore allows for bet hedging. However, this neglects the possibility of constitutive expression of σ^V . Do you see a difference in growth rate for cells with high σ^V expression levels?

A 30 minute priming time seems short compared to the switching times reported in figure 1. Could you comment on this?

The last part about 'memory' and "environmental history" (p. 10-11) sounds slightly exaggerated: If I understand it correctly, the authors suggest that the observed 'memory' effects are merely due to the dilution of σ^V operon components as the cells grow and divide. This should be toned down and ideally not be called 'memory' because this situation essentially applies to any molecule in the cell that is not actively degraded (and is certainly not reminiscent of cognitive functions). Further, these effects should be observable at the population level and some of the literature studying such effects in microbial systems should be briefly mentioned in this context.

p. 5, l. 6-18: this might be a personal preference but I would put this technicality in the methods, or limit it to 1-2 lines in the main text as it interrupts the flow.

p.7, l. 17: It should be explained in the main text that the genome-wide transcriptional response was measured by RNA-seq and a bit more detail about the experimental procedure should be added here (e.g. how long after lysozyme addition was this measurement done?).

Many of the figures show distributions, which is good practice. However, if possible, it would be nice to also show average values.

Fig. 1c. What is happening with the middle cell at 600 min? It seems that the older cells are dark and the newer cells are green.

Fig. 1e. This figure does not seem to add anything and could be removed.

Fig. 2b. Report how many cells were measured for both cases. Also, rather than fraction of survivors, could you plot a cumulative distribution of lysis times?

Point-by-Point Response

We thank the reviewers for their very useful comments and suggestions. We have carried out new experiments, the results of which further support our claims, as well as re-simulating our model with more realistic parameters. In addition, we have clarified the text in many places as suggested by the reviewers. We have also improved both the description of our RNA-seq experiment and the corresponding figure (Figure 3). We believe that the resulting text is much improved and is now suitable for publication in Molecular Systems Biology.

Reviewer #1:

In this study, the authors find that the *B. subtilis* lysozyme resistance response is activated non-uniformly across a population, in a way that depends on the stress level and environmental history. Through experiments that use varying lysozyme levels, genetic manipulations, and mathematical modeling, the authors find that this heterogeneity is mediated by a mixed positive/negative feedback network with the alternative sigma factor sigmaV at its core. This study adds to the growing body of evidence showing that gene regulation in bacteria is much more dynamic and heterogeneous than traditionally thought.

Overall, the study is well conducted, and the conclusions are supported by the experimental data. On the other hand, addressing raised technical concerns about the model and some additional experiments could further strengthen the paper conclusions.

We appreciate the reviewers' positive view of our manuscript. We have addressed their concerns below

Major Points:

R1. MP1.:

While the model is qualitatively and, in some aspects, quantitatively matches the experimental data, several technical concerns about the model ingredients make me doubt that the heterogeneity mechanism in the model is realistic. Therefore, hypothesized heterogeneity mechanism may not be the right one. Most critically, the authors use Hill equation with Hill coefficient $n=4$ to describe the gene activation by sigmaV. Given that sigma factor acts as monomeric subunit that binds to RNAP core, this cooperativity is not justified. I am certain, that both deterministic bifurcation diagram of the model showing irreversible bistability and heterogeneity of stochastic switching times that are likely based on critical slowdown resulting from this bistability critically depend on this unrealistic assumption. The authors need to verify if the model can match experimentally observed behaviors for $n=1$ and if not look for other explicit mechanism (e.g. sequestration) that are capable of producing ultrasensitive and/or bistable response in the system.

We agree with the reviewer that the Hill coefficient of 4 in the original model is not realistic. Our aim was to produce a simplified model that could describe the qualitative behaviours of the experiments. However, we did not explain this well in the original text or explain the assumptions that exist in the lumped parameters.

The reviewer is also correct that we require a Hill Coefficient greater than 1. We have remade the model using new parameters and a new simulation method (discussed below), with the results qualitatively matching the previous simulations. With the new parameters, we now have a Hill coefficient of 2. Although this is more biologically plausible than a Hill coefficient of 4, the reviewer is correct that the binding of sigma factors to polymerase is not cooperative. There are however, possible other sources of cooperativity, which we now discuss in the discussion.

This is how we address the comment in the manuscript:

Main text results:

With the exception of the production of σ^V and RsiV (which were implemented through a Hill function of σ^V activity with a Hill coefficient of 2), all reactions were modelled using mass action kinetics (See methods for further details). A Hill coefficient greater than 1 was required to generate heterogeneous activation dynamics as a degree of ultrasensitivity is needed in the system to amplify the response to molecule fluctuations.

Main text discussion:

Additionally, the system requires ultrasensitivity (in the form of a Hill coefficient n greater than 1 in the operon production term) in order to amplify molecule fluctuations. While other sigma factor response systems have been shown to utilise ultrasensitivity (Narula et al. 2012, Narula et al. 2016), there is no known source of it in the σ^V circuit (neither the binding of σ^V to RsiV, nor its operon, is cooperative). Further research should examine possible sources of ultrasensitivity in the circuit, one possibility being sigma factor competition for RNA polymerase (Park et al., 2018).

R1. MP2.:

The formalism the authors have chosen to simulate stochastic dynamics have multiple problems and need to be verified against more basic approaches, e.g. SSA simulations of the elementary reactions. First, Chemical Langevin formalism is developed for a specific types of chemical systems, in which time-steps can be selected such that a large number of reactions occur per unit time and rate functions (propensities) remain approximately constant during the same interval. The system simulated is unlikely to satisfy these assumptions (I may say it almost specifically designed to violate these). It is stiff with fast post-translational interactions (binding of sigma/anti-sigma) much faster than protein production/degradation. It has two sources of ultrasensitivity (high Hill coefficient noted above) but also large changes in free sigma when sigma_total:anti-sigma_total ratio changes for below to above 1. Second, the authors ignore bursty nature of gene expression that is widely shown to be main contributor to intrinsic noise (need to separately simulate mRNA and Proteins to properly

account for translational bursting or use well established phenomenological approximation for proper modelling of bursty expression. Third, uncorrelated noise in reaction channels assumed may not be a valid assumption. For instance, if σ / σ^V are stable and mainly degraded via dilution, the noise in the corresponding terms must be correlated but simulated independently (N_5, N_6, N_7 terms).

On the first point, we understand the reviewer's concerns about the Chemical Langevin formalism. Although we did take mitigating steps to reduce issues with stiffness in the equations, including using an implicit EM solver, which handles stiff SDEs well, in order to satisfy the reviewer concerns we have reformulated the model using a Gillespie-type reaction scheme, retaining Michaelis-Menten and Hill function terms for simplicity. We now use this model throughout the paper. Pleasingly, the dynamics that we observe match well to our previous simulations.

On the second point, we agree that we have taken a simplified assumption of noise in the system, and that it would be interesting and valid to model the noise as a bursty process, through the separation of mRNA and protein levels. Although we agree this approach to be interesting, it would not affect the qualitative conclusions that we gain from the model in this paper, and would require further experiments (e.g single molecule FISH) to correctly parameterise the bursty transcription process. We now discuss explicitly the advantages and disadvantages of our treatment of noise in the discussion, as well as discussing that in future it will be important to examine the bursty nature of expression as the reviewer suggests.

On the third point, we agree that we have used uncorrelated noise in reaction channels as a simplifying assumption. Again, we now discuss in the discussion this limitation and how in future it will be important to consider using independent noise terms, as the reviewer suggests. However, given the simplified model, our qualitative and general conclusions, and the lack of data to inform parameterising independent noise terms, we feel it is a reasonable simplification to take at this stage. We are grateful for the reviewer for pointing this out, as although we believe it is a reasonable assumption, it is important that we explain this in the text, as we now do.

This is how we address the first point in the text:

Main text:

On stress application, RsiV is degraded and σ^V is free to activate the operon. The model was simulated using Gillespie-type stochastic simulations, tracking the changes in copy-number of each species of the system by simulating the individual reaction events (Gillespie 1977).

Methods:

The model was implemented in the Julia programming language using the Catalyst.jl modelling package. Simulations were made using the DifferentialEquations.jl package (Rackauckas and Nie 2017). To account for the stochastic nature of the system, we use a Gillespie-type model (Gillespie 1977). Here, we tracked the copy-numbers of the three components of the system (σ^V , RsiV, and σ^V -RsiV), and their change due to the individual

reaction events. We used Gillespie's direct stochastic simulation algorithm to determine the time to, and which, the next reaction event in the simulation should be. We used DifferentialEquations.jl's SSAS stepper method to simulate the model. For more details, please see the implementations in the files provided.

This is how we address the second and third point:

Main text:

While simple, our model allows qualitative matches to data. In future, it will be important to increase the complexity of the model to make more precise predictions of the behaviour of the *sigV* system. One aspect of the system that can be modelled in more detail is how noise in gene expression is generated in the circuit. In our model, we do not model transcription and translation separately and assume uncorrelated noise for each reaction channel. A more detailed model could involve characterising the noise in terms of bursts of transcription and translation (Friedman et al. 2006). In turn, this would require experiments to characterise the noise in transcription, such as single-molecule FISH (Raj et al. 2008). Our assumption of uncorrelated noise is also a simplification as for example we have modelled the degradation events as uncorrelated, which may not hold as these are primarily caused by dilution.

R1. MP3.:

The main conclusion of the paper relates the phenotypic variability to the autoregulation of operon which includes *sigV*, *rsiV*, *oatA*. (with the last one being not critical for the mechanism). I am surprised the authors did not attempt to test this claim directly with replacing the autoregulated promoter of the inducible one. With a recent development of CRISPR-assisted dCas9 activation systems in bacteria such as experiment should be technically feasible.

We agree with the reviewer that it would be interesting and important to test the effects of the removal of the autoregulation of the operon. To do this, we knocked out the endogenous *sigV* locus and chromosomally integrated an IPTG inducible *sigV* operon. This allowed us to compare the behaviour of the *sigV* system at different steady-state expression levels (by varying IPTG induction level) to the WT system. We found that the fold-change induction of the WT on addition of lysozyme was at least 4.5 times higher than that observed in the inducible operon strain, regardless of the IPTG induction level. We observed the same behaviour in our model, revealing the role of the feedback loops in increasing the dynamic range of the system, allowing the amplification of underlying fluctuations. We thank the reviewer for this suggestion and feel that this new experiment improves the paper and further supports our proposed mechanism.

This is how we address the comment in the manuscript:

Main text:

Our model consists of a mixed positive and negative feedback loop. We tested the requirements of this feedback for the dynamics by modelling a feedback-broken system, with

constitutive expression of *sigV* and *rsiV*. For a range of constitutive expression, the dynamic range of P_{sigV} -YFP expression for the feedback-broken system on addition of lysozyme was significantly less than that of the WT system (Figure EV5 A and Figure S23). This reflected the role of the feedback loop in amplifying the system dynamics. To test this prediction experimentally, we constructed a strain with no autoregulation of the *sigV* operon by knocking out the *sigV* operon and replacing it with a *sigV* operon driven by an inducible promoter. This allowed us to study the system at different steady-state expression levels (by varying IPTG induction level) to the WT system. We found that the fold-change induction of the WT on addition of lysozyme was at least 4.5 times higher than that observed in the inducible operon strain, regardless of the IPTG induction level (Figure EV5 B and Figure S24), matching the behaviour observed in our model (Figure EV5 A).

Figure:

Extended View Figure 5. The *sigV* feedback loop increases the dynamic range of the circuit.

A) Sketch of WT *sigV* circuit. B) Sketch of rewired circuit with deleted positive feedback loop. C) The fold change in *sigV* expression pre/post 1 $\mu\text{g/ml}$ lysozyme stress was calculated from model simulations of the WT system (blue line) and a feedback broken system where the levels of the *sigV* operon are inducible (red line). D) The experimental fold change for the WT, and the feedback broken strain for IPTG values between 0 to 1000 μM . The blue line is the WT fold change and the red line is the fold change of $\Delta sigV rsiV oatA yrhK$ P_{spank} *sigV rsiV oatA yrhK*

Other Points:

R1. OP1:

-While it is clear that the *sigmaV* network allows cells to modulate their heterogeneity, there is not discussion about why they would want to do that instead of homogeneously switching the resistance response on. Can the authors speculate on this? Is there any previous evidence on a detrimental effect of expression of any of the *sigmaV*-regulated genes? If not, a potential fitness cost of prolonged and unnecessary *sigmaV* activation could be revealed by measuring growth from wild-type cells vs. the *sigV+* strain in the absence of lysozyme.

We agree that future work should explore more the functional significance of the heterogeneity. We now discuss this further in the discussion as future work, as well as discussing possible tests of the fitness costs of prolonged and unnecessary *sigma V* activation.

This is how we address the comment in the manuscript:

Main text:

Cells that activate *sigV* quickly after a priming stress have a higher chance of surviving a subsequent high stress (Figure 2 B). This points to a potential benefit to early activation of *sigV*. Future work should examine the costs of early *sigV* activation to see if the heterogeneous activation dynamics we observe represent a bet-hedging strategy (Veening, Stewart, et al. 2008; Veening, Smits, et al. 2008), where early responding cells suffer a fitness penalty in return for protection against future stress. We do not observe a growth rate difference in cells that activate *sigV* earlier compared to later, suggesting that early responders are not suffering a growth rate penalty. However, it is possible that we are missing small growth rate effects, as the time resolution of our mother machine experiments only allows approximately 5 time points to be measured per cell cycle. Our experiments do indicate that high constitutive expression of *sigV* or *oatA* reduces the growth rate in bulk culture (Figure S30). It could also be that the fitness penalty is due to early *sigV* activation blocking cells from responding to stress with other alternative sigma factors, as it appears alternative sigma factors compete for RNA polymerase (Park et al. 2018; Nyström 2004). Evolution experiments under changeable stressful environments could reveal whether the heterogeneity in activation and transcriptional memory that we observe evolve to optimally match the external environment.

Figure:

Figure S30 Constitutive expression of *sigV* reduces growth.

A) In the mother machine movies high levels of *sigV* expression do not seem to affect the growth rate. B) The growth curves in liquid culture when overexpressing *sigV* or *oatA* show a reduced growth rate compared to the WT. Overnight cultures were adjusted to an OD of 0.02 and grown for two hours before 1mM of IPTG was added to the cultures. The cultures were then grown for 1 h before readjusting them to an OD of 0.1 at the start of the experiment (for more details see methods).

R1. OP2:

-In Figure S6, heterogeneity in experiments conducted in an alternative microfluidic device is shown. However, this heterogeneity does not seem to decrease with increasing lysozyme concentration. Can the authors comment on this? Perhaps the experiment needed to go for longer.

The reviewer makes a good point, and actually points to the original reason why we made the shift to the mother machine from the CellAsic device. The fraction of cells that respond to the lysozyme stress increases with lysozyme in the CellAsic device, as it does in the mother machine. However, we were unable to run the movie longer in the CellAsic device to test whether all the cells eventually turn ON, as the cells filled the channel. The mother machine allowed us to run movies for much longer, avoiding the problem of cell crowding, and allowed us to observe that all cells do eventually turn ON.

So, the CellAsic experiment matches the Mother machine experiment, in the sense that only a fraction of cells immediately responds to the lysozyme stress and that this fraction increases as we increase the concentration of lysozyme. We have added a new figure panel to show this (Figure S5F). But it does not allow us to observe the system long enough to see all cells turn ON. We now explain this properly in the text and thank the reviewer for pointing this out.

This is how we address the comment in the manuscript:

Main Text:

Next, we asked whether the observed heterogeneity in P_{sigV} -YFP is modulated by the level of lysozyme applied. We examined P_{sigV} -YFP expression after the application of 0.5, 1, 2, and 4 $\mu\text{g/ml}$ lysozyme. These values were all below the previously reported minimal growth inhibitory concentration of 6.25 $\mu\text{g/ml}$ (Ho et al. 2011) and led to a transient decrease in growth rate (Figure S7). To measure the distribution of σ^V activation times, for each time point we calculated the fraction of cells that had crossed the half-maximum of their respective final σ^V value (meaning the cell had activated σ^V). We found that when increasing the lysozyme concentration from 0.5 $\mu\text{g/ml}$ to 4 $\mu\text{g/ml}$ the heterogeneity in σ^V activity was reduced (Figure 1E, Figure EV1 and Figure S8). The time for 90% of cells to activate their σ^V pathway decreased from 300 min (approximately 6 generations) for 0.5 $\mu\text{g/ml}$ to 100 min (approximately 2 generations) for 4 $\mu\text{g/ml}$ lysozyme (Figure S3). At the same time, the fold change in induction between the unstressed σ^V activity and the steady state σ^V activity under lysozyme stress increased from ~ 190 for 0.5 $\mu\text{g/ml}$ to ~ 520 for 4 $\mu\text{g/ml}$ (Figure 1F and Figure S1). We also observed that under a 4 $\mu\text{g/ml}$ concentration of lysozyme some cells (8% and 21% in two different repeat experiments) appeared sick and were wider than usual cells. These cells also overshoot their new σ^V activity steady state before relaxing to it. We removed these cells from our analysis (Figure S9), although including them did not affect our results (Figure S10). We also observed that the fraction of cells activating σ^V increased with increasing lysozyme in the alternative microfluidic device, although in this device movies were stopped before all cells activated due to crowding of the cells in the chip (Figure S5F).

Figure:

Figure S5. The observed heterogeneity in σ^V activation is not due to the geometry of the mother machine.

Cells grown in CellAsic bacteria chips also showed a heterogeneous activation of σ^V in response to lysozyme. In each subpanel stress was added after 60 min (dashed black line)

and a line corresponds to a single-cell trace. Top panel: (A) micrographs of cells grown in the CellAsic. Scale bar: 5 μm . (B) Single cell traces in response to 0 $\mu\text{g/ml}$ lysozyme. (C) Single cell traces in response to 0.5 $\mu\text{g/ml}$ lysozyme. (D) Single cell traces in response to 1 $\mu\text{g/ml}$ lysozyme. (E) Single cell traces in response to 4 $\mu\text{g/ml}$ lysozyme. (F) Histogram of single cell $P_{\text{sigV}}\text{-YFP}$ expression from in the last frame of the movie. Only a fraction of cells respond to the lysozyme stress before the end of the movie and this fraction increases with increasing concentrations of lysozyme. The plotted data is from three biological repeats.

R1. OP3:

The experiment in Figure 2 needs to be described more precisely in the main text. There, it is mentioned that the end of the movie is "280 min after the addition of 20 $\mu\text{g/ml}$ of lysozyme". However, it is not clear if this is the case for both priming and non-priming conditions, and in fact Figure S16 shows that it is not. One could think that this difference in timing could explain the observed difference in survivors. However, Figure S16 also shows that non-primed cells die almost immediately after application of 20 $\mu\text{g/ml}$ lysozyme, which is not mentioned in the main text. The diagram with no numbers in the x axis shown in Figure 2A does not help clarify this. I recommend replacing Figure 2A with the two panels in Figure S16. Furthermore, the fact that non-primed cells die immediately should be explicitly mentioned in the main text and/or the figure legend.

We agree with the reviewer that we should make it clearer that all non-primed cells die almost immediately. Reviewer 2 had a similar comment. We edited the text to make this point clearer and changed figure 2B to show the fraction of survivors.

This is how we address the comment in the manuscript:

Main text:

Given the heterogeneity in σ^{V} activation times, we examined whether activating σ^{V} early had any effect on the survival against future lethal concentrations of lysozyme. Cells that were exposed to 20 $\mu\text{g/ml}$ of lysozyme for 20 min all died within one hour (Figure 2 and Figure S15). However, if the cells were first exposed to a priming stress of 1 $\mu\text{g/ml}$ of lysozyme for 30 min, which heterogeneously induced σ^{V} , and then subsequently to 20 $\mu\text{g/ml}$ of lysozyme for 20 min, some cells survived the high lysozyme stress (Figure 2A).

Figure:

Figure 2. Rapid activation of σ^V after a first stress application increases survival after a second higher stress.

(A) Schematic of lysozyme application. Cells are either exposed directly to a high concentration of lysozyme (20 $\mu\text{g/ml}$) for 20 min (top) or exposed first to a short (30 min) lysozyme priming stress (1 $\mu\text{g/ml}$) before exposure to the higher concentration (bottom). (B) A priming (30 min) stress of 1 $\mu\text{g/ml}$ lysozyme followed by the high lysozyme stress (20 $\mu\text{g/ml}$) improves survival (No Priming: N = 2013 Priming: N = 4937). (C) Surviving cells have higher $P_{\text{sigV}}\text{-YFP}$ levels after the initial priming stress (1 $\mu\text{g/ml}$) than perishing cells. The cumulative distributions were normalized by their maximum σ^V activity and baseline subtracted. Each dashed line is the mean of experiments from n=4 biological repeats. The shaded areas correspond to the mean \pm s.d.. For more information on the number of repeats and cell numbers please see the supplementary text.

R1. OP4:

-The plots in Figures 3D and E are a little confusing. First, it is not clear why $P_{\text{sigV}}\text{-YFP}$ expression is being shown, given that we were concerned about heterogeneity in activation time and not necessarily expression. Furthermore, the main text should make it more clear when measurements were taken in relation to lysozyme and IPTG application.

We agree with the reviewer that these plots were confusing. We have changed these figures to display the fraction of activated cells rather than $P_{\text{sigV}}\text{-YFP}$ expression. We have also improved our description of when measurements were taken.

This is how we address the comment in the manuscript:

Methods:

Calculation of fraction of activated cells from Snapshots

Cells which, after the application of lysozyme stress, had a higher mean $P_{\text{sigV}}\text{-YFP}$ expression than the mean $P_{\text{sigV}}\text{-YFP}$ expression plus six standard deviations before stress were defined as having activated. The fraction of activated cells was then calculated as the number of cells which had activated normalized by the total number of cells (Figure 3 and Figure 5).

Figure:

Figure 3. The observed σ^V heterogeneity can be explained by a simplified σ^V circuit. (A) Schematic of the σ^V circuit. In the figure R (orange) is the anti-sigma factor RsiV, R* (orange) is RsiV bound to lysozyme, S (red) is signalling peptidase and RP (light blue) is RasP the site-2 protease. For more information on the activation mechanism see Figure 1B. (B & C) RNA-seq experiment on WT (JLB130) and ΔsigV (JLB154) strains, showing quantification of the expression of individual genes in the presence and absence of lysozyme stress. The shaded grey box represents a ± 5 fold change. All shown data are genes which were differentially expressed (p -value $< 5\%$) between the WT and ΔsigV mutant or in response to lysozyme treatment. (B) Only the *sigV* operon is strongly activated (>5 fold change) in response to lysozyme stress in WT (JLB130), as previously reported (Guariglia-Oropeza and Helmann 2011). (C) No genes were strongly upregulated in ΔsigV (JLB154) by lysozyme stress. (D) Effect of the overexpression of individual components of the σ^V pathway (WT; $n=8$, *sigV*+, $n=4$, *rsiV*+, $n=3$, *oatA*+, $n=4$, *yrhK*+, $n=3$, *sipS*+, $n=3$ and *rasP*+, $n=3$, *pbpX*+, $n=6$) on the fraction of activated cells. Only overexpression of *sigV*, *rsiV* or *oatA* changed the observed dynamics compared to WT. The histograms of the shown data are shown in Figure S16. (E) Deleting *oatA* did not alter the σ^V activation dynamics. However, deleting *sigV* or *rsiV* resulted in no further activation of σ^V in response to 1 $\mu\text{g/ml}$ lysozyme. $n \geq 3$ for all data shown. The histograms of the shown data are shown in Figure S17. (F) Schematic of simplified σ^V circuit with only σ^V and RsiV, where σ^V (green) activates its own expression and that of its anti-sigma RsiV (orange, R). Error bars correspond to the mean \pm s.d..

R1. OP5:

-Could the experiment in Figure 3E be conducted with *rsiV*+ too? If not, why?

We have now tested whether *rsiV* overexpression affects survival at 20 $\mu\text{g/ml}$ lysozyme, as we found for *OatA* over expression. *rsiV* overexpression shuts the *sigV* operon off, so as expected all cells die under this high lysozyme stress. In addition to this experiment, we now examine the effects of the *rsiV* mutation (as requested by reviewer 2) (Figure 3E and Figure S16). As expected from previous work, and as the model predicts, removing *rsiV* from the system causes the system to be locked into an extremely high ON state, which is not affected by the addition of lysozyme. This is because there is no mechanism to capture *sigV*, and so it activates the operon to extremely high levels, irrespective of lysozyme. We feel the addition of these experiments further increases our understanding of the system.

This is how we address the comment in the manuscript:

Main text:

Finally, overexpressing *oatA*, which blocks lysozyme cleavage of the peptidoglycan, shuts off the activation of σ^V (Figure 3D and Figure S16). However, when we increased the lysozyme concentration to 20 $\mu\text{g/ml}$ in the presence of *oatA* overexpression, the heterogeneous expression of P_{sigV} -YFP reappeared (Figure 3 and Figure EV3), suggesting *oatA* is not responsible for the heterogeneous activation dynamics. We repeated the experiment for *rsiV* overexpression, but increased RsiV did not protect against lysozyme as a lethal concentration of 20 $\mu\text{g/ml}$ lysozyme killed all cells.

To further validate the importance of *sigV* and *rsiV* as compared to *oatA* for the observed heterogeneity in σ^V activation, we constructed deletion mutants of *sigV*, *rsiV* and *oatA*. Only the deletion of *oatA* left the activation σ^V dynamics unchanged (Figure 3E). In the *sigV* mutant, P_{sigV} -YFP levels did not increase in response to lysozyme stress (Figure 3C and Figure S17). Deleting *rsiV* caused all cells to have high P_{sigV} -YFP expression even before the addition of lysozyme and the addition of lysozyme did not activate the system any further (Figure S17). These findings suggest that the heterogeneity in σ^V activation only depends on σ^V and its anti-sigma factor RsiV (Figure 3).

Figure:

Figure 3. The observed σ^V heterogeneity can be explained by a simplified σ^V circuit. (A) Schematic of the σ^V circuit. In the figure R (orange) is the anti-sigma factor RsiV, R* (orange) is RsiV bound to lysozyme, S (red) is signalling peptidase and RP (light blue) is RasP the site-2 protease. For more information on the activation mechanism see Figure 1B. (B & C) RNA-seq experiment on WT (JLB130) and ΔsigV (JLB154) strains, showing quantification of the expression of individual genes in the presence and absence of lysozyme stress. The shaded grey box represents a ± 5 fold change. All shown data are genes which were differentially expressed (p -value $< 5\%$) between the WT and ΔsigV mutant or in response to lysozyme treatment. (B) Only the sigV operon is strongly activated (>5 fold change) in response to lysozyme stress in WT (JLB130), as previously reported (Guariglia-Oropeza and Helmann 2011). (C) No genes were strongly upregulated in ΔsigV (JLB154) by lysozyme stress. (D) Effect of the overexpression of individual components of the σ^V pathway (WT; $n=8$, $\text{sigV}+$; $n=4$, $\text{rsiV}+$; $n=3$, $\text{oatA}+$; $n=4$, $\text{yrhK}+$; $n=3$, $\text{sipS}+$; $n=3$ and $\text{rasP}+$; $n=3$, $\text{pbpX}+$; $n=6$) on the fraction of activated cells. Only overexpression of sigV , rsiV or oatA changed the observed dynamics compared to WT. The histograms of the shown data are shown in Figure S16. (E) Deleting oatA did not alter the σ^V activation dynamics. However, deleting sigV or rsiV resulted in no further activation of σ^V in response to 1 $\mu\text{g/ml}$ lysozyme. $n \geq 3$ for all data shown. The histograms of the shown data are shown in Figure S17. (F) Schematic of simplified σ^V circuit with only σ^V and RsiV, where σ^V (green) activates its own expression and that of its anti-sigma RsiV (orange, R). Error bars correspond to the mean \pm s.d..

Figure S17. Deletion of *oatA* does not change the induction dynamics of P_{sigV} -YFP. The first column corresponds to 0 µg/ml lysozyme and the second column corresponds to 1 µg/ml lysozyme.

Each row is a different deletion mutant: (A) WT (JLB130), (B) $\Delta sigV$ (JLB154), (C) $\Delta rsiV$ (JLB208), (D) $\Delta oatA$ (JLB156).

Reviewer #2:

Summary

The authors investigate the stress response of *B. Subtilis* to lysozyme, an enzyme that is produced by the immune system to kill bacteria via cell wall lysis. The genetic components involved in this stress response are well known. The authors leverage this knowledge to investigate how the genetic architecture governs the cell response. They start by measuring the expression levels of a key regulator, σV , in response to lysozyme stress. Importantly, going beyond previous work, they measure the single-cell diversity by using state-of-the-art microfluidics (the mother machine). They find a highly heterogeneous stress response, especially at low stress. Further, exposure to low concentrations of lysozyme 'primes' cells by increasing the level of σV ; this protects cells from subsequent higher doses of lysozyme. Based on RNA-seq data and a $\Delta\sigma V$ mutant they suggest that no other pathways are involved in the stress response. Using multiple fluorescent reporter genes, they next dissect the σV circuit and conclude that only two components, σV and $RsiV$ are responsible for the single-cell heterogeneity. A model of this minimal circuit of two components can indeed reproduce key characteristics of the single-cell heterogeneity (stochastic switching, increased heterogeneity at low stress and a stronger switch at higher stress). Lastly, they show that the σV circuit can 'memorize' previous stresses as elevated σV levels only gradually decay upon removal of lysozyme stress.

General remarks

The study is well motivated, well described and, technically, seems largely thorough. The topic is relevant for microbiologists interested in lysozyme stress and alternative sigma factors, and more broadly for systems biologists and biophysicists interested in the effects of phenotypic heterogeneity. While this work does not provide new insight into molecular interactions, it provides a rationale for the architecture of the σV regulatory network. Heterogeneous stress responses via discrete (meta-)stable states have been shown before, but how cells generate and control such heterogeneity using genetic circuits is rarely understood. This work makes progress in this direction. However, I have several concerns regarding the interpretation of the experimental data, the mathematical model, and the design of some of the experiments, which need to be addressed.

We appreciate the reviewer's positive assessment of our work. We also thank them for their careful examination of the paper and have addressed their concerns below, through a combination of new experiments, improved modelling, and re-writing of the text.

Major points:

R2. MP1.:

The effects of changes in mean expression level and expression heterogeneity need to be disentangled more clearly. The problem is that any change in mean expression level will generally affect heterogeneity, with higher expression levels leading to lower heterogeneity in relative terms (quantified e.g. by the variation coefficient). This has been established in various genome-wide studies (see e.g. Fig. 1F in PMID 22275871, Fig. 2B in PMID 20671182, Fig. 2a in PMID 16715097). This phenomenon needs to be taken into account when interpreting experiments in which the system is perturbed and the mean expression level and its variability change. This affects the interpretation of several experiments but it is particularly important for conclusions as in l. 7-8 on p. 10: given the studies above, it is clear that heterogeneity in expression can be tuned by genetic changes that affect the mean expression level. It would be a much more convincing point if the authors can show that this tuning can be done without changing mean expression level (or if the effects go far beyond those expected from changes in the mean alone).

We agree with the reviewer that several genome-wide studies have shown that mean expression level and heterogeneity in a distribution are linked. The heterogeneity we are focused on in the paper is the distribution of turn ON times after stress, rather than a static distribution. This is most easily observed in the mother machine experiments, as we have time-lapse data so can observe the state of the cell through the turn ON process, but in the static snapshot data we now focus on 'Fraction of cells activated' rather than the mean and CV, to avoid the problems that the reviewer points out. We thank the reviewer for pointing out this issue. We also explain that the perturbations are also affecting the final mean levels of the distributions, and include the histograms of the snapshots of gene expression in the supplementary files.

This how we address the comment in the text:

Methods:

Calculation of fraction of activated cells from Snapshots

Cells which, after the application of lysozyme stress, had a higher mean P_{sigV} -YFP expression than the mean P_{sigV} -YFP expression plus six standard deviations before stress were defined as having activated. The fraction of activated cells was then calculated as the number of cells which had activated normalized by the total number of cells (Figure 3 and Figure 5).

Figure:

Figure 3. The observed σ^V heterogeneity can be explained by a simplified σ^V circuit. (A) Schematic of the σ^V circuit. In the figure R (orange) is the anti-sigma factor RsiV, R* (orange) is RsiV bound to lysozyme, S (red) is signalling peptidase and RP (light blue) is RasP the site-2 protease. For more information on the activation mechanism see Figure 1B. (B & C) RNA-seq experiment on WT (JLB130) and $\Delta sigV$ (JLB154) strains, showing quantification of the expression of individual genes in the presence and absence of lysozyme stress. The shaded grey box represents a ± 5 fold change. All shown data are genes which were differentially expressed (p -value $< 5\%$) between the WT and $\Delta sigV$ mutant or in response to lysozyme treatment. (B) Only the *sigV* operon is strongly activated (>5 fold change) in response to lysozyme stress in WT (JLB130), as previously reported (Guariglia-Oropeza and Helmann 2011). (C) No genes were strongly upregulated in $\Delta sigV$ (JLB154) by lysozyme stress. (D) Effect of the overexpression of individual components of the σ^V pathway (WT; $n=8$, *sigV* $+$; $n=4$, *rsiV* $+$; $n=3$ *oatA* $+$; $n=4$, *yrhK* $+$; $n=3$, *sipS* $+$; $n=3$ and *rasP* $+$; $n=3$, *pbpX* $+$; $n=6$) on the fraction of activated cells. Only overexpression of *sigV*, *rsiV* or *oatA* changed the observed dynamics compared to WT. The histograms of the shown data are shown in Figure S16. (E) Deleting *oatA* did not alter the σ^V activation dynamics. However, deleting *sigV* or *rsiV* resulted in no further activation of σ^V in response to 1 $\mu\text{g/ml}$ lysozyme. $n \geq 3$ for all data shown. The histograms of the shown data are shown in Figure S17. (F) Schematic of simplified σ^V circuit with only σ^V and RsiV, where σ^V (green) activates its own expression and that of its anti-sigma RsiV (orange, R). Error bars correspond to the mean \pm s.d..

R2. MP2.:

It is unclear if overexpression (and deletion) of circuit components (p. 7, l. 24-25) can really disentangle which components contribute to the heterogeneity of the response. These are drastic perturbations, which affect mean expression levels (see previous point). Even if they don't, it is not obvious what we can conclude from an effect of overexpressing component A on the expression variability of component B without also quantifying the variability of component A (and correlating it with that of B). The rationale for this needs to be explained in more detail.

We agree that full overexpression and deletion of the circuit components are quite drastic perturbations. We used these perturbations to test which components were involved in the heterogeneous activation of *sigV*. What was convincing to us was that although these perturbations are quite drastic, for *yrhK*, *rasP*, and *sipS*, and *oatA* these perturbations had no effect on the heterogeneous activation of *sigV*. This gave us confidence to focus on *sigV* and *rsiV* for the further experiments and for the model. To understand the effects of *sigV* and *rsiV* we agree that smaller perturbations are required. This is the reason for us then carrying out experiments examining the effects of adding a second copy of *sigV*, *rsiV*, and *sigV-rsiV* components (Figure S26). We now also include observations of the effects of the leakiness of the inducible operon with no IPTG (Figure S18 and Figure S19). We have re-written the text to acknowledge the issues with the larger perturbations, as well as explaining the effects of the smaller perturbations.

This is how we address the comment in the manuscript:

Main text:

We found that the leakiness of the inducible *sigV* construct ($P_{hyperspank^-} sSigV$, without addition of IPTG) increased the fraction of activated cells on the addition of lysozyme, as well as causing an increase in the steady state levels of $P_{sigV} yfp$ before and after stress. The leakiness of the *RsiV* construct ($P_{hyperspank^-} rRsiV$) caused the opposite effect. These effects were still apparent for inducible constructs with lower leakiness ($P_{spank^-} sigV$, $P_{spank} rsiV$ (Figure S18 and Figure S19)), confirming that *sigV* activation dynamics are sensitive to these two system components.

Figure:

Figure S18. The $sigV$ activation dynamics are sensitive to $sigV$ and $rsiV$ baseline levels.

A) The baseline expression of $P_{spank}\text{-YFP}$ is almost the same as a strain without a YFP reporter. Baseline expression of blank (JLB54), $P_{hyperspank}\text{-YFP}$ (JLB244) and $P_{spank}\text{-YFP}$ (JLB242) with 0 μM IPTG are shown. B&C) The $sigV$ activation dynamics are sensitive to σ^V and $RsiV$ baseline levels. The histograms for blank (JLB54), WT (JLB130), $P_{spank}\text{-sigV}$ (JLB209), $P_{spank}\text{-rsiV}$ (JLB196), $P_{hyperspank}\text{-sigV}$ (JLB219) and $P_{hyperspank}\text{-rsiV}$ (JLB193) are shown for 0 $\mu\text{g/ml}$ lysozyme (B) and 1 $\mu\text{g/ml}$ Lysozyme (C).

Figure S19. The $sigV$ activation dynamics are sensitive to σ^V and RsiV baseline levels.

The first column corresponds to 0 $\mu\text{g/ml}$ lysozyme and the second column corresponds to 1 $\mu\text{g/ml}$ lysozyme. Each row is a different mutant: (A) blank (JLB54), (B) WT (JLB130), (C) $P_{spank}\text{-sigV}$ (JLB209), (D) $P_{spank}\text{-rsiV}$ (JLB196), (E) $P_{hyperspank}\text{-sigV}$ (JLB219), (F) $P_{hyperspank}\text{-rsiV}$ (JLB193).

Figure S26. Snapshots confirm that the σ^V response to lysozyme is genetically tunable.

The first column corresponds to 0 µg/ml lysozyme and the second column corresponds to 1 µg/ml lysozyme. Each row is a different mutant: (A) WT (JLB130), (B) 2x *sigV* (JLB210), (C) 2x *rsiV* (JLB212), (D) 2x *sigVrsiV* (JLB211).

R2. MP3.:

Using this approach, the authors find that the expression level of three genes is affected by lysozyme stress (σ^V , *RsiV* and *oatA*) but conclude that *oatA* is not involved in the heterogeneity of the stress response, as σ^V expression is still heterogeneous at elevated levels of lysozyme when *oatA* is overexpressed. Therefore, they conclude that a minimal network to explain heterogeneity consists of σ^V and *RsiV*. However, no check is performed with an *RsiV*+ mutant. To investigate whether *RsiV* is indeed part of the minimal circuit needed to explain heterogeneity, a similar check should be done for this protein as for *oatA* in Fig 3E.

We agree with the reviewer that it's important to examine the effects of the *rsiV* mutation. We have carried out this experiment, and the results behave consistent with expectations from the literature and the model. We have added an additional figures to show this (Figure 3E and Figure S17).

This is how we address the comment in the manuscript:

Main text:

Finally, overexpressing *oatA*, which blocks lysozyme cleavage of the peptidoglycan, shuts off the activation of σ^V (Figure 3D and Figure S16). However, when we increased the lysozyme concentration to 20 µg/ml in the presence of *oatA* overexpression, the heterogeneous expression of P_{sigV} -YFP reappeared (Figure EV3), suggesting *oatA* is not responsible for the heterogeneous activation dynamics. We repeated the experiment for *rsiV* overexpression, but increased *RsiV* did not protect against lysozyme as a lethal concentration of 20 µg/ml lysozyme killed all cells.

To further validate the importance of *sigV* and *rsiV* as compared to *oatA* for the observed heterogeneity in σ^V activation, we constructed deletion mutants of *sigV*, *rsiV* and *oatA*. Only the deletion of *oatA* left the activation σ^V dynamics unchanged (Figure 3E). In the *sigV* mutant, P_{sigV} -YFP levels did not increase in response to lysozyme stress (Figure 3C and Figure S17). Deleting *rsiV* caused all cells to have high P_{sigV} -YFP expression even before the addition of lysozyme and the addition of lysozyme did not activate the system any further (Figure S17). These findings suggest that the heterogeneity in σ^V activation only depends on σ^V and its anti-sigma factor *RsiV* (Figure 3).

Figure:

Figure 3. The observed σ^V heterogeneity can be explained by a simplified σ^V circuit. (A) Schematic of the σ^V circuit. In the figure R (orange) is the anti-sigma factor RsiV, R* (orange) is RsiV bound to lysozyme, S (red) is signalling peptidase and RP (light blue) is RasP the site-2 protease. For more information on the activation mechanism see Figure 1B. (B & C) RNA-seq experiment on WT (JLB130) and $\Delta sigV$ (JLB154) strains, showing quantification of the expression of individual genes in the presence and absence of lysozyme stress. The shaded grey box represents a ± 5 fold change. All shown data are genes which were differentially expressed (p -value $< 5\%$) between the WT and $\Delta sigV$ mutant or in response to lysozyme treatment. (B) Only the *sigV* operon is strongly activated (>5 fold change) in response to lysozyme stress in WT (JLB130), as previously reported (Guariglia-Oropeza and Helmann 2011). (C) No genes were strongly upregulated in $\Delta sigV$ (JLB154) by lysozyme stress. (D) Effect of the overexpression of individual components of the σ^V pathway (WT; $n=8$, *sigV* $+$; $n=4$, *rsiV* $+$; $n=3$ *oatA* $+$; $n=4$, *yrhK* $+$; $n=3$, *sipS* $+$; $n=3$ and *rasP* $+$; $n=3$, *pbpX* $+$; $n=6$) on the fraction of activated cells. Only overexpression of *sigV*, *rsiV* or *oatA* changed the observed dynamics compared to WT. The histograms of the shown data are shown in Figure S16. (E) Deleting *oatA* did not alter the σ^V activation dynamics. However, deleting *sigV* or *rsiV* resulted in no further activation of σ^V in response to 1 $\mu\text{g/ml}$ lysozyme. $n \geq 3$ for all data shown. The histograms of the shown data are shown in Figure S17. (F) Schematic of simplified σ^V circuit with only σ^V and RsiV, where σ^V (green) activates its own expression and that of its anti-sigma RsiV (orange, R). Error bars correspond to the mean \pm s.d..

Figure S17. Deletion of *oatA* does not change the induction dynamics of P_{sigV} -YFP. The first column corresponds to 0 µg/ml lysozyme and the second column corresponds to 1 µg/ml lysozyme.

Each row is a different deletion mutant: (A) WT (JLB130), (B) $\Delta sigV$ (JLB154), (C) $\Delta rsiV$ (JLB208), (D) $\Delta oatA$ (JLB156).

R2. MP4.:

The verification part of the model (page 9) is not very convincing. Basically, the authors show that the mean time and standard deviation scale with each other. However, this is not only true for this specific genetic network, but for just about any model that considers stochastic transition events. The authors should at least mention that these predictions are not particularly unique for this specific model/network motif. To verify the model further, the fit parameters should be explained in more detail. How many free parameters are there? Are they constrained by the data? Are their values plausible based on literature knowledge? Since it is a stochastic model, it should also be possible to comment on the copy numbers of the circuit components in the model, which should be compared to experimental estimates.

We agree with the reviewer that the description of the model and its verification should be improved. We have revised the model based on Reviewer 1's comments and now discuss its limitations in the discussion. The model does make qualitative matches to data beyond just showing that the mean time and standard deviation scale with each other. These include matching the effects of perturbations to the levels of σ^V and RsiV, the new match between experiment and model for the effects of removing the feedbacks on the system, as well as predicting the transcriptional memory that we observe in the system. We now verify our model predictions further, by justifying our parameter choices and commenting on the copy number of *sigV* molecules in the text, as suggested by the reviewer, as well as testing how sensitive our simulations are to a +/-10% shift in parameter values (Figure S21).

This is how we address the comment in the manuscript:

Main text results 1:

The *sigV* operon components were assumed to be stable, so the dilution degradation rate was set to approximately match the division rate observed in experiments. Simulations yielded plausible copy numbers for the number of sigma factor molecules (Jishage and Ishihama 1995, Jishage et al. 1996). In addition, we verified that the heterogeneous activation behaviour is robust to perturbations to the system parameters (Figure S21).

Main text results 2:

Finally, the heterogeneous activation dynamics were also modulated by small increases in the baseline production rate of either σ^V or RsiV (Figure S22), qualitatively matching the effects of the leakiness of the inducible σ^V or RsiV construct observed in experiment (Figure S18).

Main text results 3:

Our model consists of a mixed positive and negative feedback loop. We tested the requirements of this feedback for the dynamics by modelling a feedback-broken system, with constitutive expression of *sigV* and *rsiV*. For a range of constitutive expression, the dynamic range of P_{sigV} -YFP expression for the feedback-broken system on addition of lysozyme was significantly less than that of the WT system (Figure EV5 A and Figure S23). This reflected the role of the feedback loop in amplifying the system dynamics. To test this prediction experimentally, we constructed a strain with no autoregulation of the *sigV* operon by knocking out the *sigV* operon and replacing it with a *sigV* operon driven by an inducible promoter. This allowed us to study the system at different steady state expression levels (by varying IPTG induction level) to the WT system. We found that the fold-change induction of the WT on addition of lysozyme was at least 4.5 times higher than that observed in the inducible operon strain, regardless of the IPTG induction level (Figure EV5 B and Figure S24), matching the behaviour observed in our model (Figure EV5 A).

Main text discussion:

While simple, our model allows qualitative matches to data. In future, it will be important to increase the complexity of the model to make more precise predictions of the behaviour of the *sigV* system. One aspect of the system that can be modelled in more detail is how noise in gene expression is generated in the circuit. In our model, we do not model transcription and translation separately and assume uncorrelated noise for each reaction channel. A more detailed model could involve characterising the noise in terms of bursts of transcription and translation (Friedman et al. 2006). In turn, this would require experiments to characterise the noise in transcription, such as single-molecule FISH (Raj et al. 2008). Our assumption of uncorrelated noise is also a simplification as for example we have modelled the degradation events as uncorrelated, which may not hold as these are primarily caused by dilution. Additionally, the system requires ultrasensitivity (in the form of a Hill coefficient n greater than 1 in the operon production term) in order to amplify molecule fluctuations. While other sigma factor response systems have been shown to utilise ultrasensitivity (Narula et al. 2012, Narula et al. 2016), there is no known source of it in the σ^V circuit (neither the binding of σ^V to RsiV, nor its operon, is cooperative). Further research should examine possible sources of ultrasensitivity in the circuit, one possibility being sigma factor competition for RNA polymerase (Park et al., 2018).

Figure:

Figure S21. The model behaviour is preserved under parameter perturbations.

(A-H) Each parameter of the system is perturbed by 10%, and the effect on the cumulative activation diagram is shown (the parameter L is not perturbed, since this is equivalent to perturbing kC). Perturbing K has a relatively large effect on the system, however, the normal activation behaviour is still exhibited, as seen in (I), which contains 15 trajectories for the activation of a wildtype system (blue), a system where K has been reduced by 10% (red) and increased by 10% (green). The heterogeneous activation behaviour is preserved, and the parameter perturbation simply tunes the activation.

R2. MP5.:

This work needs to be better put into the context of the field. For several points, recent single-cell studies on other microbial stress responses made conceptually similar observations. For example, an increase of gene expression variability when an inducer (or stressor) concentration is decreased (p. 6, l. 10-12) is often observed for gene expression responses, see e.g. Fig. 2 in PMID 18469087. A conceptually similar single-cell experiment about 'priming' bacteria before a higher stress is added was recently done in PMID 28342718 and could be mentioned here.

We agree with the reviewer that we should cite these important studies and put the work in better context in the field. We have updated the introduction and discussion accordingly in order to do this.

This is how we address the comment in the manuscript (New references are highlighted in yellow):

Introduction:

Cells live in a changeable environment and experience a wide range of environmental stresses. Bacterial populations have evolved strategies to survive these stresses. One strategy is for all cells to immediately respond to stress with the activation of the relevant stress response pathway (Hilker et al. 2016). Alternatively, a bacterial population can generate a broad range of cellular states, which allows it to hedge its bets against the changeable environment (Veening, Stewart, et al. 2008). Noise in gene expression has been proposed as a mechanism for generating phenotypic variability in genetically identical cells (Raj and van Oudenaarden 2008; Martins and Locke 2015). This phenotypic variability has also been shown to be affected by changes in the cellular environment, such as a shift in stress level or growth conditions (Mitosch et al. 2019; de Jong et al. 2012; Megerle et al. 2008), as well as by previous 'priming' stresses (Mitosch et al. 2017). However, how the bacterial population regulates individual cell decisions in order to modulate the fraction of cells that enter an alternative transcriptional state remains unclear (Figure 1A).

Discussion:

Our modelling and experiments found that recent exposure to lysozyme stress modulates the heterogeneity observed on a second stress application, even though the system turns off after removal of the first lysozyme stress. The key to this behaviour appears to be the 'mixed' feedback loop, as it allows amplified levels of inactive σ^V -RsiV complexes in each cell after stress. These complexes can be cleaved by a subsequent addition of lysozyme, releasing σ^V to activate its operon. Similar transcriptional memories of previous stress have been observed in bacterial systems, although typically not to modulate phenotypic heterogeneity. For example, other pathways such as the heat stress response in *B. subtilis* (Runde et al. 2014) or the oxidative stress response in yeast (Kelley and Ideker 2009) have been shown to have a transcriptional memory of past conditions. Often this transcriptional memory is facilitated through an autoregulatory positive feedback loop that can lock the cell into an ON state that is heritable through cell divisions (Lambert and Kussell 2014; Novick and Weiner 1957; Acar et al. 2005; Hashimoto et al. 2013; Biggar and Crabtree 2001; Xiong and Ferrell 2007). However, it is difficult for a single positive feedback loop to allow the system to be OFF but primed for future stress, as we find to for the 'mixed' feedback loop in the σ^V pathway. In the case of σ^V dilution during growth causes heterogeneous activation of *sigV* to eventually return when levels of σ^V -RsiV drop to pre-stress levels, so the memory is qualitatively different from that generated by a positive feedback loop locking a system ON. However, we find that the *sigV* transcriptional memory remains for several generations. In the future, it will be important to investigate whether the 'mixed' feedback loop also tunes the levels of phenotypic diversity by environmental history in other systems. Interestingly, computational studies have proposed that a 'mixed' feedback loop structure in the MAR operon in *E. coli* allows the tuning of the fraction of cells prepared to survive antibiotic exposure (Garcia-Bernardo and Dunlop 2013). Additionally the mixed feedback loop mechanism could be compared to other mechanisms proposed to allow the modulation of

phenotypic variability, such as multi-site phosphorylation (Libby et al. 2019) or threshold-based mechanisms in toxin-antitoxin modules (Rotem et al. 2010).

While simple, our model allows qualitative matches to data. In future, it will be important to increase the complexity of the model to make more precise predictions of the behaviour of the *sigV* system. One aspect of the system that can be modelled in more detail is how noise in gene expression is generated in the circuit. In our model, we do not model transcription and translation separately and assume uncorrelated noise for each reaction channel. A more detailed model could involve characterising the noise in terms of bursts of transcription and translation (Friedman et al. 2006). In turn, this would require experiments to characterise the noise in transcription, such as single-molecule FISH (Raj et al. 2008). Our assumption of uncorrelated noise is also a simplification as for example we have modelled the degradation events as uncorrelated, which may not hold as these are primarily caused by dilution. Additionally, the system requires ultrasensitivity (in the form of a Hill coefficient n greater than 1 in the operon production term) in order to amplify molecule fluctuations. While other sigma factor response systems have been shown to utilise ultrasensitivity (Narula et al. 2012, Narula et al. 2016), there is no known source of it in the σ^V circuit (neither the binding of σ^V to RsiV, nor its operon, is cooperative). Further research should examine possible sources of ultrasensitivity in the circuit, one possibility being sigma factor competition for RNA polymerase (Park et al., 2018).

Other points:

R2. OP1:

Showing a dose response curve (growth rate vs. [lysozyme]) would be helpful to put the used concentrations into context.

We agree that this plot would be useful and we have added a dose response curve to the supplementary material (Figure S7)

This is how we address the comment in the manuscript:

Main text:

Next, we asked whether the observed heterogeneity in P_{sigV} -YFP is modulated by the level of lysozyme applied. We examined P_{sigV} -YFP expression after application of 0.5, 1, 2, and 4 $\mu\text{g/ml}$ lysozyme. These values were all below the previously reported minimal growth inhibitory concentration of 6.25 $\mu\text{g/ml}$ (Ho et al. 2011) and led to a transient decrease in growth rate (Figure S7).

Figure:

Figure S7. Growth rate transiently decreases after the application of stress, with the minimal growth rate during the growth rate dip decreasing with increasing stress levels.

(A) Single-cell growth rate traces for 0, 0.5, 1, 2 and 4 µg/ml lysozyme stress. The stress was added after 240 min at the black dashed line. After a spike in measured growth rate on application of stress (which could be due to changes in cell wall properties leading to an apparent cell size increase), growth rate transiently decreases. (B) Plot of minimal growth rate vs. lysozyme concentration.

R2. OP2:

The authors show that high levels of σ^V protect cells against lysozyme exposure, and that a heterogeneous cell response therefore allows for bet hedging. However, this neglects the possibility of constitutive expression of σ^V . Do you see a difference in growth rate for cells with high σ^V expression levels?

In our microfluidic experiments we do not see a difference in growth rate for high *sigV* expression levels (Figure S30A). This might be due to the low imaging interval of 5 images per cell cycle. When we constitutively expressed *sigV* and *oatA* and took the growth curves (Figure S30 B) we did observe that high *sigV* or *oatA* expression reduced the growth rate. We now discuss this in the discussion.

This is how we address the comment in the manuscript:

Main text:

Cells that activate *sigV* quickly after a priming stress have a higher chance of surviving a subsequent high stress (Figure 2 B). This points to a potential benefit to early activation of *sigV*. Future work should examine the costs of early *sigV* activation to see if the heterogeneous activation dynamics we observe represent a bet-hedging strategy (Veening, Stewart, et al. 2008; Veening, Smits, et al. 2008), where early responding cells suffer a

fitness penalty in return for protection against future stress. We do not observe a growth rate difference in cells that activate *sigV* earlier compared to later, suggesting that early responders are not suffering a growth rate penalty. However, it is possible that we are missing small growth rate effects, as the time resolution of our mother machine experiments only allows approximately 5 time points to be measured per cell cycle. Our experiments do indicate that high constitutive expression of *sigV* or *oatA* reduces the growth rate in bulk culture (Figure S30). It could also be that the fitness penalty is due to early *sigV* activation blocking cells from responding to stress with other alternative sigma factors, as it appears alternative sigma factors compete for RNA polymerase (Park et al. 2018; Nyström 2004). Evolution experiments under changeable stressful environments could reveal whether the heterogeneity in activation and transcriptional memory that we observe evolve to optimally match the external environment.

Figure:

Figure S30 Constitutive expression of *sigV* reduces growth.

A) In the mother machine movies high levels of *sigV* expression do not seem to affect the growth rate. B) The growth curves in liquid culture when overexpressing *sigV* or *oatA* show a reduced growth rate compared to the WT. Overnight cultures were adjusted to an OD of 0.02 and grown for two hours before 1mM of IPTG was added to the cultures. The cultures were then grown for 1 h before readjusting them to an OD of 0.1 at the start of the experiment (for more details see methods).

R2. OP3:

A 30 minute priming time seems short compared to the switching times reported in figure 1. Could you comment on this?

We agree that the use of a 30 minute priming stress is short compared to the time taken for all cells to switch. Although chosen somewhat arbitrarily, we used a 30 minute priming stress to ensure the heterogeneous activation of *sigV*, with a large fraction of cells not having turned on before the addition of the second higher stress. We now mention this heterogeneity in the text.

This is how we address the comment in the manuscript:

Main text:

Given the heterogeneity in σ^V activation times, we examined whether activating σ^V early had any effect on the survival against future lethal concentrations of lysozyme. Cells that were exposed to 20 $\mu\text{g/ml}$ of lysozyme for 20 min all died within one hour (Figure 2 and Figure S15). However, if the cells were first exposed to a priming stress of 1 $\mu\text{g/ml}$ of lysozyme for 30 min, which heterogeneously induced σ^V , and then subsequently to 20 $\mu\text{g/ml}$ of lysozyme for 20 min, some cells survived the high lysozyme stress (Figure 2A). We chose this priming stress level and duration as previous experiments had shown (Figure S4) that it ensured heterogenous activation of σ^V , with a large fraction of cells not having turned on before the second higher stress.

R2. OP4:

The last part about 'memory' and "environmental history" (p. 10-11) sounds slightly exaggerated: If I understand it correctly, the authors suggest that the observed 'memory' effects are merely due to the dilution of *sigV* operon components as the cells grow and divide. This should be toned down and ideally not be called 'memory' because this situation essentially applies to any molecule in the cell that is not actively degraded (and is certainly not reminiscent of cognitive functions). Further, these effects should be observable at the population level and some of the literature studying such effects in microbial systems should be briefly mentioned in this context.

We understand the reviewer's concerns about our use of the term 'memory' in the text. We now refer to 'transcriptional memory', to reduce comparison to cognitive function. Although we agree with the reviewer that the *sigV* transcriptional memory requires that the *sigV* operon components are not actively degraded, it is different to the situation for any molecule in the cell due to the fact that the memory is caused by the system being trapped in an OFF state, ready to respond to immediately respond to future stresses. However, we agree that the *sigV* component memory is not as stable as those caused by locking a system ON through positive feedback for example. We have added a paragraph to the discussion to discuss these points and put the work into better context.

This is how we address the comment in the manuscript:

Main text:

Our modelling and experiments found that recent exposure to lysozyme stress modulates the heterogeneity observed on a second stress application, even though the system turns off after removal of the first lysozyme stress. The key to this behaviour appears to be the 'mixed' feedback loop, as it allows amplified levels of inactive σ^V -RsiV complexes in each cell after stress. These complexes can be cleaved by a subsequent addition of lysozyme, releasing σ^V to activate its operon. Similar transcriptional memories of previous stress have been observed in bacterial systems, although typically not to modulate phenotypic heterogeneity. For example, other pathways such as the heat stress response in *B. subtilis* (Runde et al. 2014) or the oxidative stress response in yeast (Kelley and Ideker 2009) have

been shown to have a transcriptional memory of past conditions. Often this transcriptional memory is facilitated through an autoregulatory positive feedback loop that can lock the cell into an ON state that is heritable through cell divisions (Lambert and Kussell 2014; Novick and Weiner 1957; Acar et al. 2005; Hashimoto et al. 2013; Biggar and Crabtree 2001; Xiong and Ferrell 2007). However, it is difficult for a single positive feedback loop to allow the system to be OFF but primed for future stress, as we find to for the 'mixed' feedback loop in the σ^V pathway. In the case of σ^V , dilution during growth causes heterogeneous activation of *sigV* to eventually return when levels of σ^V -RsiV drop to pre-stress levels, so the memory is qualitatively different from that generated by a positive feedback loop locking a system ON. However, we find that the *sigV* transcriptional memory remains for several generations.

R2. OP5:

p. 5, l. 6-18: this might be a personal preference but I would put this technicality in the methods, or limit it to 1-2 lines in the main text as it interrupts the flow.

We agree that the flow is disrupted by these sentences, but would prefer to keep them in the main text as evidence for heterogeneous activation in liquid culture and an alternative microfluidic device are quite important for the generality of our claims.

R2. OP6:

p.7, l. 17: It should be explained in the main text that the genome-wide transcriptional response was measured by RNA-seq and a bit more detail about the experimental procedure should be added here (e.g. how long after lysozyme addition was this measurement done?).

We added more information on the RNA-seq experiment to the text. In particular, that the RNA-seq experiment was done 30 min after the addition of lysozyme and how we decided on the stress strength and duration. In addition, we now also discuss *pbpX*, a lysozyme resistance gene in the *sigV* regulon.

This is how we address the comment in the manuscript:

Main text:

We next attempted to understand how the single-cell activation dynamics of the σ^V pathway are generated. First, to test whether the heterogeneity that we observed in σ^V activation times was due to lysozyme stress activating different stress response pathways, we analysed the genome-wide transcriptional response of cells to 1 $\mu\text{g/ml}$ lysozyme. We carried out RNA-seq 30 min after the addition of stress in the wild type and in the $\Delta\textit{sigV}$ knockout. We chose this stress level and duration as we had seen in previous experiments that it resulted in heterogeneous σ^V activation (Figure S4). As previously reported, only the *sigV* operon was strongly (>5 fold induction) induced by lysozyme (Guariglia-Oropeza and Helmann 2011) in the wild type (Figure 3B). The lysozyme resistance gene *pbpX*, which is part of the *sigV* regulon, was also upregulated in the WT (4.9 fold induction), but was not

upregulated in the $\Delta sigV$ background (0.88 fold induction), consistent with the known role of *sigV* in its activation (Guariglia-Oropeza and Helmann 2011).

Methods:

Lysozyme treatment before RNA extraction

Overnight cultures were prepared as for snap shot or movie experiments. Briefly, cultures were started from frozen stock in SMM and grown overnight at 30°C to an OD between 0.3 and 0.8. The overnight cultures were resuspended to an OD of 0.01 and regrown to an OD of 0.1 at 37°C. Once the cultures had reached an OD of 0.1, they were split into aliquots of 10 ml to which either 0 µg/ml or 1 µg/ml lysozyme from hen eggs white (Sigma Aldrich, St. Louis, MO, USA) were added. The aliquots were then incubated at 37°C for 30 min before snap freezing them.

R2. OP7:

Many of the figures show distributions, which is good practice. However, if possible, it would be nice to also show average values.

We have added mean and CV values to the histograms of single-cell snapshots of gene expression.

R2. OP8:

Fig. 1c. What is happening with the middle cell at 600 min? It seems that the older cells are dark and the newer cells are green.

This is due to the fact that cells further down the channel ("newer cell") can also switch on *sigV*. At 600 min the cell in the middle of the chip switches on, whilst the mother cell does not. We also see the reverse, where the mother cell switches but the other cells have not switched.

R2. OP9:

Fig. 1e. This figure does not seem to add anything and could be removed.

Readers of the draft manuscript found this figure helpful to visualize the variability in activation times. We would like to keep this figure for this purpose, although we will remove it if the editors or reviewers wish.

R2. OP10:

Fig. 2b. Report how many cells were measured for both cases. Also, rather than fraction of survivors, could you plot a cumulative distribution of lysis times?

We have changed Figure 2B to show the fraction of surviving cells during the course of the movie (in order to satisfy reviewer 1's concerns). We also include the total number of cells for each experimental condition in the caption. In addition to this, we note that we include a supplementary table with details of the number of repeats and N for each figure.

This is how we address the comment in the manuscript:

Figure:

Figure 2. Rapid activation of σ^V after a first stress application increases survival after a second higher stress.

(A) Schematic of lysozyme application. Cells are either exposed directly to a high concentration of lysozyme (20 µg/ml) for 20 min (top) or exposed first to a short (30 min) lysozyme priming stress (1 µg/ml) before exposure to the higher concentration (bottom). (B) A priming (30 min) stress of 1 µg/ml lysozyme followed by the high lysozyme stress (20 µg/ml) improves survival. (No Priming: N = 2013, Priming: N = 4937). (C) Surviving cells have higher P_{sigV} -YFP levels after the initial priming stress (1 µg/ml) than perishing cells. The cumulative distributions were normalized by their maximum σ^V activity and baseline subtracted. Each dashed line is the mean of experiments from n=4 biological repeats. The shaded areas correspond to the mean \pm s.d.. For more information on the number of repeats and cell numbers please see the supplementary text.

Thank you for sending us your revised manuscript. We have now heard back from the two reviewers who were asked to evaluate your study. As you will see, the reviewers are overall satisfied with the modifications made and think that the study is now suitable for publication.

Before we can formally accept your manuscript, we would ask you to address the following editorial-level issues.

REFEREE REPORTS

Reviewer #1:

While I am not fully convinced with the author's responses to my technical comments regarding the Hill Coefficient >1 and treatment of the gene expression noise, I think the developed models are sufficient for the stated goals of qualitatively explaining trends in the data. Therefore, I am OK with the publication of the revised manuscript.

Reviewer #2:

The authors have convincingly addressed all major issues I had raised in my previous report. They have revised significant parts of the main text and added new experimental data, which strengthens this work. In particular, the addition of the rsiV overexpression experiment is helpful. Importantly, this work is now much better placed in the context of the relevant literature. I am still slightly worried that the terms "memory" or "transcriptional memory" used to describe the reported phenomenon may be misleading or at least confusing for some readers, but the explicit comparison to a positive feedback loop in the Discussion alleviates this concern. Overall, I am happy to support publication of this work.

2nd Authors' Response to Reviewers

28th May 2021

The authors have made all requested editorial changes.

Thank you again for sending us your revised manuscript. We are now satisfied with the modifications made and I am pleased to inform you that your paper has been accepted for publication.

Corresponding Author Name: James Locke

Manuscript Number: MSB-20-9832